# A projectome of the bumblebee central complex

**Marcel Ethan Sayre[1,2]\*, Rachel Templin[3†], Johanna Chavez[1], Julian Kempenaers[1], Stanley Heinze[1,4]\***

[1]Lund University, Lund Vision Group, Department of Biology, Lund, Sweden; [2]Macquarie University, Department of Biological Sciences, Sydney, Australia; [3]Queensland Brain Institute, University of Queensland, Brisbane, Sweden; [4]Lund University, NanoLund, Lund, Sweden

**Abstract** Insects have evolved diverse and remarkable strategies for navigating in various ecologies all over the world. Regardless of species, insects share the presence of a group of morphologically conserved neuropils known collectively as the central complex (CX). The CX is a navigational center, involved in sensory integration and coordinated motor activity. Despite the fact that our understanding of navigational behavior comes predominantly from ants and bees, most of what we know about the underlying neural circuitry of such behavior comes from work in fruit flies. Here, we aim to close this gap, by providing the first comprehensive map of all major columnar neurons and their projection patterns in the CX of a bee. We find numerous components of the circuit that appear to be highly conserved between the fly and the bee, but also highlight several key differences which are likely to have important functional ramifications.

**\*For correspondence:**
marcel.sayre@biol.lu.se (MES);
stanley.heinze@biol.lu.se (SH)

**Present address:** [†]European Molecular Biology Laboratory (EMBL), Cell Biology and Biophysics Unit, Heidelberg, Germany

**Competing interests:** The authors declare that no competing interests exist.

## Introduction

Honeybees and desert ants are iconic navigators (Reviewed in *Collett, 2019*). Honeybees can forage tens of kilometers away from their nest and can also communicate the distance and direction of foraging locations to fellow workers (*Frisch, 1967*). Desert ants forage hundreds of meters away from their nest in landscapes sometimes entirely bereft of visual landmarks (*Wehner, 2020*). Both insects preferentially make use of the most reliable visual cues, such as polarized skylight, but can also make use of an arsenal of backups to ensure a robust compass signal, including windflow (*Müller and Wehner, 2007*) and magnetoreception (*Collett and Baron, 1994*; *Fleischmann et al., 2020*).

Although their navigational behavior is less well characterized than that of honeybees and desert ants, bumblebees are also impressive foragers, as well as generally charismatic insects that play vital ecological roles (*Goulson, 2003*). Bumblebees forage hundreds to thousands of meters away from their nest and have been shown to be capable of homing from novel locations at distances up to 9.8 km (*Goulson and Stout, 2001*; *Prŷs-Jones and Corbet, 2011*). Like honeybees, bumblebees rely on celestial cues such as polarized light for homing (*Wellington, 1974*) and have been shown to compensate for wind drift (*Riley et al., 1999*). Bumblebees also show a remarkable capacity to learn novel tasks (*Loukola et al., 2017*; *Chittka, 2017*).

While behaviorally, navigation and orientation are best understood in ants and bees, our understanding of how insect brains control navigation behavior has been largely obtained in a set of different species, most notably the desert locust *Schistocerca gregaria*, the fruit fly *Drosophila melanogaster* and more recently, Monarch butterflies (*Danaus plexippus*) and dung beetles (*Scarabaeus sp.*; Reviewed in *Honkanen et al., 2019*; *Turner-Evans and Jayaraman, 2016*). Yet none of these species show the sophisticated homing behavior of hymenopteran insects and the question remains how these tiny brains enable such flexible and impressive navigational skills.

**eLife digest** Bumblebees forage widely for pollen and nectar from flowers, sometimes travelling kilometers away from their nest, but they can somehow always find their way home in a nearly straight line. These insects have been known to return to their nest from new locations almost 10 kilometers away. This homing ability is a complex neurological feat and requires the brain to combine several processes, including observing the external world, controlling bodily movements and drawing on memory.

While the navigational behavior of bees has been well-studied, the neuronal circuitry behind it has not. Unfortunately, most of what is known about insects' brain activity comes from studies in species such as locusts or fruit flies. In these species, a region of the brain known as the central complex has been shown to have an essential role in homing behaviors. However, it is unknown how similar the central complex of bumblebees might be to fruit flies' or locusts', or how these differences may affect navigational abilities.

Sayre et al. obtained images of thin slices of the bumblebee central complex using a technique called block-face electron microscopy, which produces high-resolution image volumes. These images were used to obtain a three-dimensional map of over 1300 neurons. This cellular atlas showed that key aspects of the central complex are nearly identical between flies and bumblebees, including the internal compass that monitors what direction the insect is travelling in. However, hundreds of millions of years of independent evolution have resulted in some differences. These were found in neurons possibly involved in forming memories of the directions and lengths of travelled paths, and in the circuits that use such vector memories to steer the insects towards their targets. Sayre et al. propose that these changes underlie bees' impressive ability to navigate.

These results help explain how the structure of insects' brains can determine homing abilities. The insights gained could be used to develop efficient autonomous navigation systems, which are challenging to build and require a lot more processing power than offered by a small part of an insect brain.

Work in other insects has revealed that one region of the insect brain lies at the heart of navigational control: the central complex (CX; *Honkanen et al., 2019*). The CX is highly conserved across insects and plays important functional roles in multi-sensory integration, coordinated motor activity, and sleep. The CX receives external and internal sensory information that informs the insect about its visual environment, mechanosensory cues, and self-motion cues (*Seelig and Jayaraman, 2013*; *el Jundi et al., 2014*; *Okubo et al., 2020*). CX cells use these cues to generate sensory maps that encode the heading direction (*Heinze and Homberg, 2007*; *Seelig and Jayaraman, 2015*; *Varga and Ritzmann, 2016*) and traveling direction (*Lu et al., 2020*; *Lyu et al., 2020*; *Hulse et al., 2020*). The CX also likely compares current heading with a desired goal direction to generate appropriate motor commands (*Stone et al., 2017*; *Rayshubskiy et al., 2020*; *Hulse et al., 2020*). A function of the CX in the context of motor control has indeed been firmly established (*Martin et al., 2015*; *Pfeiffer and Homberg, 2014*).

Across insects, the CX comprises four distinct neuropils (five in the fruit fly *Wolff and Rubin, 2018*): the protocerebral bridge (PB), fan-shaped body (FB; also called the upper division of the central body; CBU), the ellipsoid body (EB; lower division of the central body; CBL), and the noduli (NO). These neuropils are highly interconnected by the arborizations of three classes of neurons. First, columnar cells are central to most computations carried out by the CX. They link the four adjacent CX neuropils and provide the basis for intrinsic computations, but also generate output to other brain areas. Input to the CX moves predominantly via the second class, tangential neurons, which connect many different brain regions to distinct compartments of the CX. The third class comprise pontine cells, including hΔ cells and, in flies, vΔ cells (*Hulse et al., 2020*). These are FB interneurons whose input and output processes are confined to the FB. For each class of cell, there exist numerous cell types, each of which is defined by their detailed morphology, polarity, and ultimately, by their connectivity to other neurons.

One of the keys to understanding how this brain region is involved in so many different processes, ranging from motor control, sensory encoding, to sleep control, is the tight relation between

structure and function, facilitated by the almost crystalline anatomical layout of this brain area. This link has inspired much research on the CX, which has culminated in the recent release of the first ever comprehensive map of all CX neurons and their connectivity, achieved for the fruit fly CX (*Scheffer et al., 2020*). Even in this previously highly studied and well-understood model organism, connectomics analysis revealed the presence of many new cell types and produced novel insights into neural pathways and computational circuit motifs (*Hulse et al., 2020*).

Yet, the question arises how much of this *Drosophila* connectomics data is representative for other species and how much of it is specific for the fruit fly and its ecology. Neuroanatomical and physiological work outside of the fruit fly has clearly revealed differences in the anatomy and physiology in the CX between species (*Heinze and Homberg, 2008*; *Heinze et al., 2013*; *Stone et al., 2017*; *El Jundi et al., 2018*; *Hensgen et al., 2021*; *von Hadeln et al., 2020*). However, the anatomical data from all species except the fruit fly is generally based on sparse labeling methods for identifying neurons, mostly intracellular dye injections and immunohistochemistry. While these methods yield beautiful morphological features of individual neurons, they are not comprehensive and often biased toward the largest neurons. Importantly, these studies will never be able to answer which neural elements are missing, no matter how many neurons are reported, and can neither provide definite data on cell quantities, projectivity motifs and connectivity.

To address this issue, a counterpart of the *Drosophila* CX connectome will be needed. Obtaining similar data in a bee species will have the power to validate computational principles identified in the fly but also expand our understanding of how more complex behavioral abilities, that is, those not found in flies, are controlled by the CX.

Here, we provide the first step in such an endeavour by reporting the first comprehensive map of neural projections in the bumblebee CX (i.e. 'projectome'; *Kasthuri and Lichtman, 2007*). Using cellular resolution serial block-face electron microscopy (SBEM), we traced the main neurites of all major CX columnar cells and FB interneurons, amounting to over 1300 neural skeletons. From these reconstructions, we established information about the numbers of cells, cell type identity and their projection patterns. Due to the tight structure function links in the CX, this detailed account of projection patterns yields a first approximation of the information flow principles in the bee CX. We additionally identified several cell types that may be unique to the bee, providing starting points for more detailed investigations. Most importantly, our findings highlight a conserved core circuitry of the CX and thus contribute to the notion that the CX contains an ancestral circuit, highly preserved across insects, but with additional layers or modifications adapted to species-specific ecologies and behaviors.

## Results

In total, we reconstructed more than 1300 columnar and hΔ neurons across three data sets from three individuals that varied in resolution (Figure 2a). The lowest resolution data set (126 nm) was used to trace the majority of all CX neurons. However, due to the relatively coarse resolution of 126 nm, we were not able to resolve cell types with main axon diameters of less than 0.6 µm, so that an unknown number of neuron types with small fiber diameters are missing from our analysis. This precluded tracing of most tangential cells, in particular in the EB, PB, and FB.

To aid our estimation of total cell quantities, we additionally traced two higher resolution data sets from different individual bees. These were both centered on the NO, with one containing within it a partial CX (100 nm resolution), while the second one covered a single nodulus (24 nm resolution; Figure 2a). Both of these data sets had already been used for the SBEM analyses presented in *Stone et al., 2017*. Due to these additional higher resolution image stacks, the most detailed data are available for the NO. Additionally, we characterized CX neuropils using immunohistochemical staining and obtained full morphologies of certain individual neurons using intracellular dye injections.

Although our projectome analysis was limited to the large neurons with columnar arborizations, these reconstructions provide the most comprehensive neuroanatomical survey to date of CX cell types in any insect apart from the fruit fly.

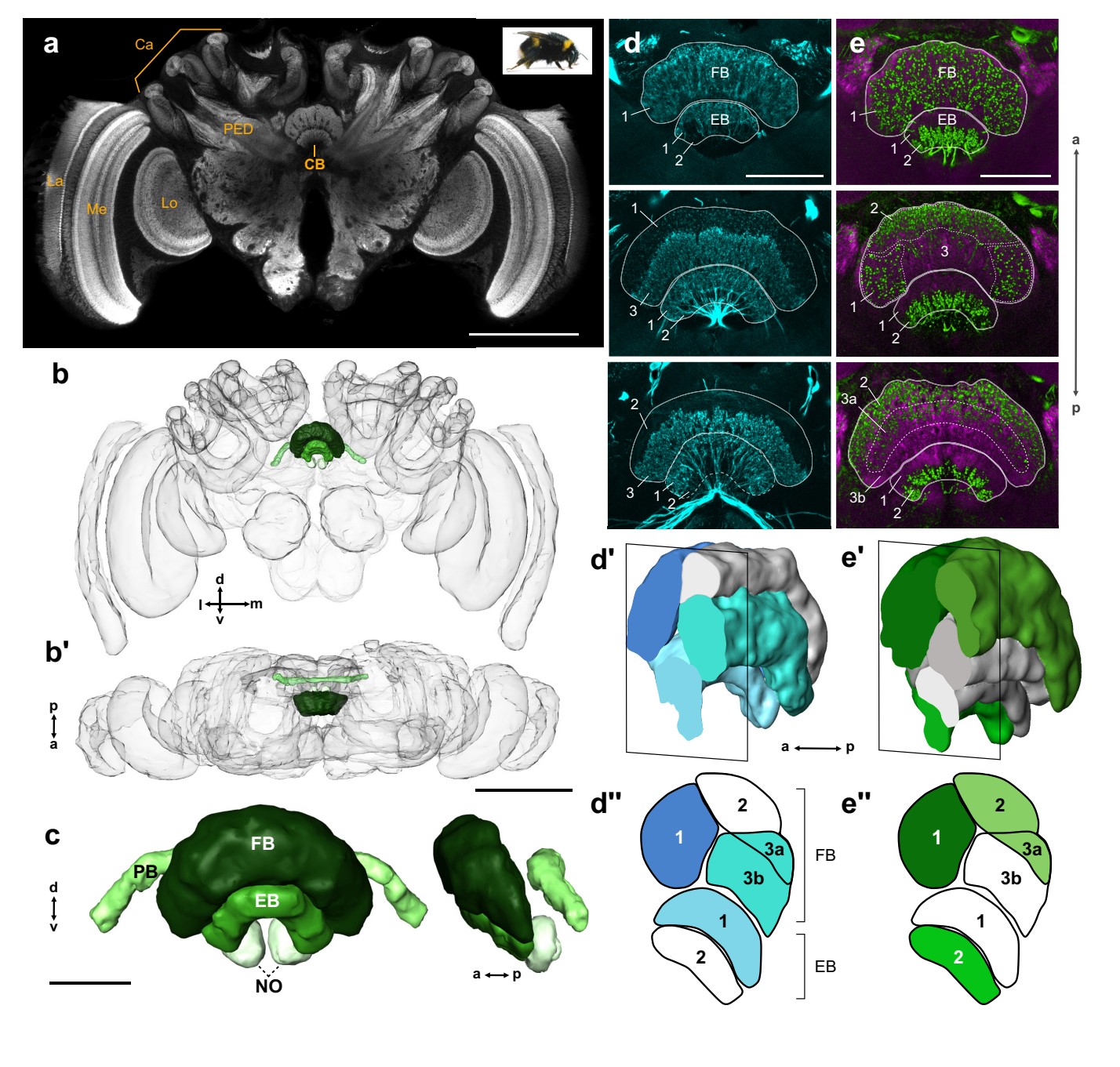

**Figure 1.** General structure of the bumblebee central complex. (**a**) Synapsin immunolabeling of a bumblebee brain. (**b–b'**) 3D reconstruction of entire bee brain with CX in shades of green, oriented frontally (**b**) and horizontally (**b'**). (**c**) Frontal and lateral views of a segmented bumblebee CX. (**d**) Whole brain TH immunolabeling in the CB from anterior (top image) to posterior (bottom image). Numbered layers correspond to (**d''**). (**d'**) Reconstruction of TH immunoreactive layers (blue) registered to the CX from 126 nm data set (see Materials and methods). Due to the lack of synapsin co-labeling in TH immunolabeled brains, the lower border of the EB was not identifiable and layer 1 was therefore not reconstructed. (**d''**) Schematic of (**d'**). (**e–e''**) Same as (**d–d''**), but for serotonin (green) and synapsin (magenta) immunolabeling. La, lamina; Me, medulla; Lo, lobula; Ca, calyx; PED, pedunculus; CB, central body. Scale bars: (**a–b'**) 500 μm, (**c–e**) 100 μm.

## General structure of the bumblebee CX

As in other insects, the bumblebee CX spans the mid-line of the protocerebrum in the central brain and is composed of four adjacent neuropils (*Figure 1a–c*), when labeled with antibodies against the

presynaptic marker synapsin (*Klagges et al., 1996*). In order from posterior to anterior, these are the protocerebral bridge (PB), the paired noduli (NO), the fan-shaped body (FB; alternatively named upper division of the central body), and the ellipsoid body (EB; alternatively named lower division of the central body) (*Figure 1c*). The FB lays over the top of the EB and together they form the central body (CB).

As no internal structure was visible based on anti-synapsin labeling alone, we used labeling against neurotransmitters to highlight further sub-compartments. Immunohistochemical staining with antibodies raised against serotonin (5HT) and tyrosine hydroxylase (TH) revealed horizontal layering in the CB (*Figure 1d–e''*). In preparations labeled with anti-TH, at least three layers can be identified in the FB; one that contains very fine processes, another with blebbed fibers, and a third lacking any TH immunoreactivity (*Figure 1d*). In samples stained against 5HT, a similar three-layer pattern emerged, containing zones with either fine branches, blebbed fibers, or no immunoreactivity at all (*Figure 1e*). The EB contained at least two layers, a dorsal layer filled with fibers from dopaminergic (TH) neurons and a ventral layer innervated by serotonergic processes.

To see how the observed layering in the CB corresponded across preparations and to electron microscopy data, whole brains stained with either TH or 5HT were imaged, manually segmented, and registered to the surface mesh of the 126 nm SBEM data set, which was used as reference volume (*Figure 1d',e'*; see Materials and methods). The resulting registrations revealed at least four distinct layers in the FB, two of which were TH and 5HT immunoreactive (layers 1 and 3a; *Figure 1d'',e''*), one which was only occupied by TH fibers (layer 3b; *Figure 1d''*), and a fourth that appeared to only contain 5HT-positive processes (layer 2; *Figure 1e''*).

Interestingly, the PB was devoid of both TH- and 5HT-positive processes and synapsin immuno-labeling did not reveal any glomeruli-like structure as is the case in the fly (*Wolff et al., 2015*). Only two compartments could be discerned in the NO, one small and clearly demarked in synapsin stained tissue termed here NOs, and the other large, containing a coarse mesh of beaded, seroto-nergic processes (NOm; see Results further on regarding the NO).

Based on projection patterns of individual columnar neurons, the CX was further divided into ver-tical columns (discussed further in the next section). However, no indication of columns was evident based on immunohistochemical labeling.

## Columnar cell projection patterns

All columnar cells originated from cell bodies situated dorsally of the CX. Their fibers extended ven-trally to the PB where they branched to make contacts with their first synaptic partners. Following the PB, columnar cells fasciculated into prominent bundles. Eight in total, these bundles spanned the width of the CX and were bilaterally symmetric about the mid-line with four bundles in each hemisphere. The resulting formation was that of a chiasm, where neurons from a bundle in one hemi-sphere projected in the direction of the contralateral hemisphere (*Figure 2b*). From lateral to medial, these bundles are generally referred to as the W, X, Y, and Z bundles (*Williams, 1975*) and can be associated with either the right or left hemisphere with the addition of L or R abbreviation (i.e. RW or LZ; *Figure 2b,h*). In locusts and flies these bundles correspond to the developmental origin of a neuron from one of four neuroblasts on either side of the midline (*Boyan and Reichert, 2011*; *Boyan et al., 2017*), and therefore they served as our prime reference when describing columnar arborization patterns.

Within the PB, each of these bundles generally gave rise to two neighboring columnar arboriza-tion domains. An exception to this rule was the layout of the innermost bundle (Z-bundle), in which traced neurons innervated three adjacent arborization domains in each hemisphere (*Figure 2b*). Therefore, our data demonstrated that the bumblebee PB contains 18 columns (nine in each hemi-sphere), which are connected to nine corresponding columns in the CB (*Figure 2c,d*). Columns in the PB were numbered L1/R1 to L9/R9 starting at the midline, while columns of the CB were num-bered C1-C9 from right to left (*Figure 2d*). Upon reaching the CB, columnar cells defasciculated from their bundle and branched into discrete zones, usually, but not always, within a well-defined column. Finally, except for one cell type discussed below, columnar cells projected further, linking the PB and CB with the NO or another tertiary neuropil outside of the CX.

We next determined the innervated columns for all neurons of a given bundle, i.e. all neurons in the identity-giving W, X, Y, and Z bundles. All columnar cells followed one of four projection pat-terns (*Figure 2e,f*). Columnar cell types that followed the 'default' projection pattern had no offset

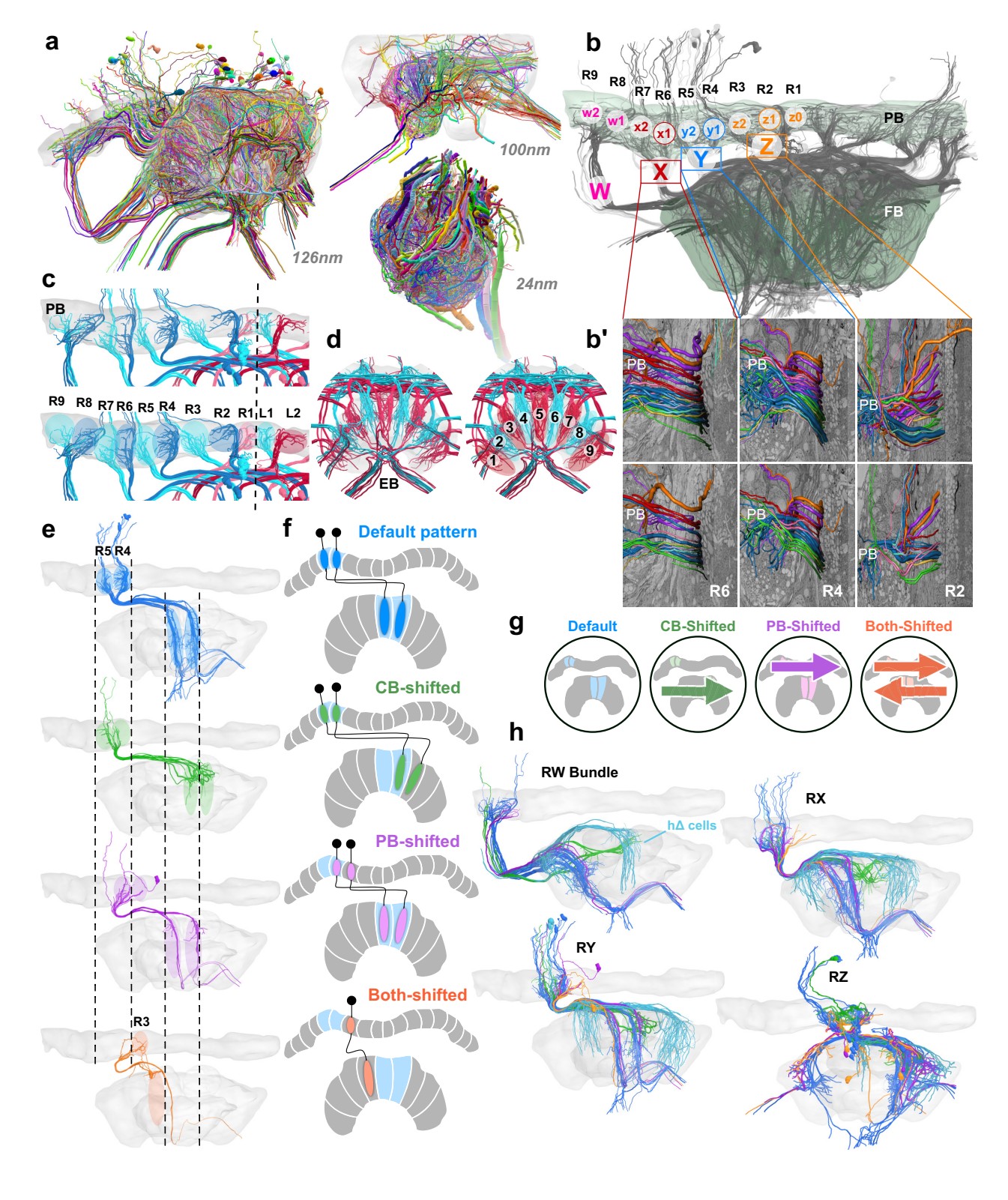

**Figure 2.** Columnar cell projection patterns in the bumblebee CX. (**a**) Views of the three SBEM data sets used in this study. (**b**) Horizontal view of 126 nm data set showing W, X, Y, Z bundles and their associated columns in the PB. (**b'**) Columnar cells are ordered from dorsal to ventral by projection type within bundle. (**c–d**) Nine structural columns in the bee CX; snapshots of EPG/PEG arborization domains in the PB (**c**) and EB (**d**). (**e**) Horizontal view of the four columnar projection patterns that characterize all CX columnar neurons in the bee. (**f**) Schematics of the four projection patterns. *Figure 2 continued on next page*

*Figure 2 continued*

Names of each projection pattern is given by the arborization of the cell relative to the bundle it projects from (see also Figure 13). (**g**) Icons for each of these patterns that are paired with cell types in the remaining figures to aid with identification. (**h**) Horizontal view of columnar and hΔ cells (light blue) projecting from each bundle in the right CX hemisphere.

in the CB relative to the bundle they originate from. That is, neurons of the lateralmost PB-column (L9/R9) connected to the lateralmost CB column (CB1/CB9) and neurons from each of the following columns innervated a CB column shifted medially by one column. Thus, all neurons of one PB hemisphere projected to corresponding regions covering the entire width of the CB. Interestingly, neurons of the medial-most columns of the PB (L1/R1) are swapped between the hemispheres (*Figure 2c*), creating an interlocked layout of the neuronal systems that belong to either hemisphere.

Three alternative projection patterns were observed that could be derived from the described default pattern by shifting either the column of origin in the PB or by shifting the target projection in the CB (*Figure 2e,f*). Cells which are 'CB-shifted' were shifted by one CB column in the direction of the contralateral hemisphere, whereas cells that are 'PB-shifted' were shifted by one PB column toward the ipsilateral hemisphere. Finally, cells labeled here as 'both-shifted' were shifted by one PB column in the direction of the contralateral hemisphere and one CB column in the ipsilateral hemisphere (*Figure 2f*). In these alternative projection patterns, neurons did not exist in all PB columns. Interestingly, while most neurons followed the default pattern, all cell types that projected to the NO were characterized by the CB-shifted pattern, while neurons that are likely to be main CX output neurons *Stone et al., 2017*; *Rayshubskiy et al., 2020*; *Hulse et al., 2020* followed a pattern with shifted PB innervation.

## Columnar cell types

We identified columnar cell types based on their projection fields within CX neuropils, their heterolateral projection patterns in the CB (i.e. the column they project to relative to the bundle they originate from), their neurite paths, their population size (number of cells per PB column), and by the tertiary neuropil they innervate (*Figure 3*). Overall, we found eleven columnar cell types, two that innervated the EB (*Figure 3a*) and nine which innervated the FB (*Figure 3b*). For a list of corresponding neuron names historically used in other insects, see *Table 1*. For additional information on nomenclature and the neuron naming scheme, see Methods. Quantities of each cell type per PB column can be found in *Figure 3—figure supplement 1*.

Columnar cell types projecting to the EB included EPG/PEG cells and PEN cells (*Figure 3a*). EPG/PEG cells arborized in the PB, the EB, and the gall , while PENs sent projections through the PB, the EB, and a small compartment in the NO (NOs; Figure 6a–c).

All FB columnar cells contained processes that arborized in both the PB and the FB. Several FB cell types sent projections toward the lateral complex (LX). These included PFL1,3, PFL2, PFLx, and PFx2 cells (*Figure 3b*). PFx1 cells sent their fibers away from the CX toward regions in the inferior protocerebrum (INP) and lateral accessory lobes (LAL), while PFx2 had descending fibers toward the LX, and PFx3 and PFx4 neurons contained fibers that extended anteriorly toward regions in the superior medial protocerebrum (SMP) and INP. Lastly, one cell type, PF1, contained arborizations in only the PB and FB. The morphology, quantity and projections of these cells are discussed further in the sections that follow.

## The noduli: PFN and PEN cells

In flying insects, the paired NO are two spherical units situated ventrally in the CX, posterior to the CB (*Figure 1c*). In bees and flies, the NO receive mixed arborizations from PFN (*Figure 4a–e*) and PEN cells (*Figure 4d–e*), which propagate signals to and from the CB and PB columns in the contralateral hemisphere, as well as input from large tangential cells (*Figure 4f*; *Stone et al., 2017*; *Hensgen et al., 2021*; *von Hadeln et al., 2020*; *Hulse et al., 2020*).

To reveal the detailed cellular architecture of the bumblebee NO, we traced all input neurons as well as all columnar neurons that supply the NO in our high-resolution data (24 nm resolution). Based on these analyses, the projections from tangential neurons delineated three domains in the

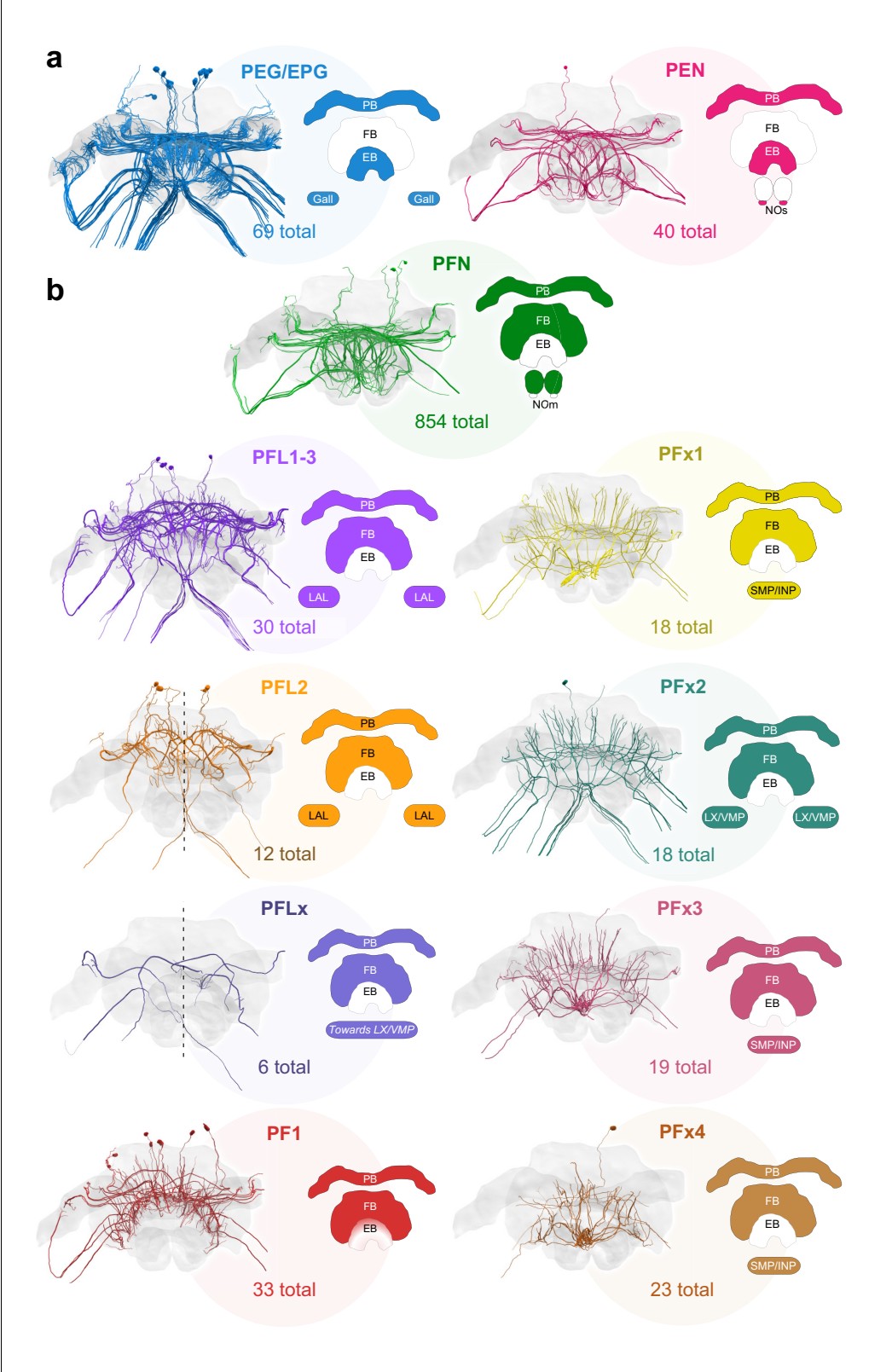

**Figure 3.** Major identified columnar cell types of the bumblebee CX. (**a–b**) Frontal views of each cell type. Colored regions in CX schematics indicate neuropil arborization domains for each cell type. The number underneath each cell type represents the total number of traceable cells for each cell type. Cell number totals are based on tracing data from the 126 nm data set with the exception of PFN and PEN cells, the numbers of which were derived from the 24 nm data set (see *Figure 3—figure supplement 1*). (**a**) Columnar cells that innervate the EB. (**b**) Columnar cells that innervate the

*Figure 3 continued*

FB. Due to resolution and field of view limits of our data sets, the PFN total is likely an underestimate. Numbers for PFx4 neurons are likely an underestimate as well, as the main axon diameter of this cell type was close to the resolution limit of our image data. Note also that confidence for correct identification of smaller cells is lower in the W-bundle compared to the other bundles. See *Table 1* for corresponding neuron names previously used in other insects. PB, protocerebral bridge; FB, fan-shaped body; EB, ellipsoid body; NO, noduli; LX, lateral complex; LAL, lateral accessory lobe; INP, inferior protocerebrum; SMP, superior medial protocerebrum; VMP, ventromedial protocerebrum.

The online version of this article includes the following figure supplement(s) for figure 3:

**Figure supplement 1.** Distribution of columnar cell types per PB column.

bumblebee NO: the small unit (NOs), main unit (NOm), and the cap (NOc; *Figure 4f–g*). These regions were supplied by distinct types of tangential cells. Two small types (LNO4 and LNO5) innervated specifically the NOs, while two larger neurons densely innervated the entire NOm with numerous fibers (LNO1 and LNO2; *Figure 4f*). A fifth large neuron, entering the NO together with the LNO4/5 cells, innervated a subcompartment of the NOm overlapping with the projection fields of LNO1/2 cells (LNO3). Finally, a set of 15–20 smaller input cells projected to the NOc region. These cells did not have neurites projecting from the direction of the ipsilateral lateral accessory lobes (LAL), but emerged from a fiber bundle that vertically passed the EB and originated in the contralateral anterior protocerebrum (*Figure 4d*). The same bundle contained numerous tangential cells of the FB, suggesting that the tangential neurons supplying the NOc region also belong to this group of neurons. This proposed morphology was confirmed by an intracellular dye fill from *Megalopta* (*Figure 4—figure supplement 1d*) and we named these neurons FB-NOc cells.

With respect to columnar cells our reconstructions revealed 12–20 PFN neurons with large to medium fiber diameters in the NOm, while PEN cells were restricted to the NOs (*Figure 4d–e,g*). Additional cells with smaller fiber diameters originated alongside the larger PFN neurons, and mostly projected exclusively to the NOc region. Those cells which projected to the NOc were termed PFNc cells and their collective fiber bundles were adjacent to the main PFN cells along the entire neurite path toward the PB (*Figure 4e*). They thus most likely constitute the only identifiable subtype of PFN cells. Given their projection fields, these cells overlapped only with the FB-NOc cells but not with LNO neurons (*Figure 4e–f*; *Figure 4—figure supplement 1a–b*).

Compared to other columnar cells of the CX, PFN cells are by far the most numerous cell type. In our low resolution data, we identified 202 PFN cells total, while in the 100 nm data we identified a

**Table 1.** List of corresponding neuropil and neuron names in *Drosophila melanogaster* and other insects.

Names of neurons from other species are based on locust nomenclature (*Heinze and Homberg, 2008*), and have been used in butterflies (*Heinze et al., 2013*), moths (*de Vries et al., 2017*), beetles (*El Jundi et al., 2018*), and bees (*Stone et al., 2017*; *Hensgen et al., 2021*). Asterisks indicate cells identified in this study with no clear homologues in *Drosophila*.

| CX neuropils | | Columnar neurons | | FB interneurons | | Tangential neurons | |
|---|---|---|---|---|---|---|---|
| *Drosophila* | Other insects | *Drosophila* | Other insects | *Drosophila* | Other insects | *Drosophila* | Other insects |
| PB | PB | EPG/PEG | CL1a/b | hΔ | Pontines | Δ7 | TB1 |
| FB | CBU | PEN | CL2 | vΔ | *Unknown* | FBx | TU |
| EB | CBL | PFL1 | CPU1 Type 1 | | | ER | TL |
| NO | NO | PFL3 | CPU1 Type 2 | | | LNO | TN |
| | | PFL2 | CPU2 | | | | |
| | | PFN | CPU4 | | | | |
| | | PFR | *Unknown* | | | | |
| | | PFG | *Unknown* | | | | |
| | | FX | CU | | | | |
| | | *Unknown* | PF1* | | | | |
| | | *Unkown* | PFx1,2,3,4* | | | | |
| | | *Unknown* | PFLx* | | | | |

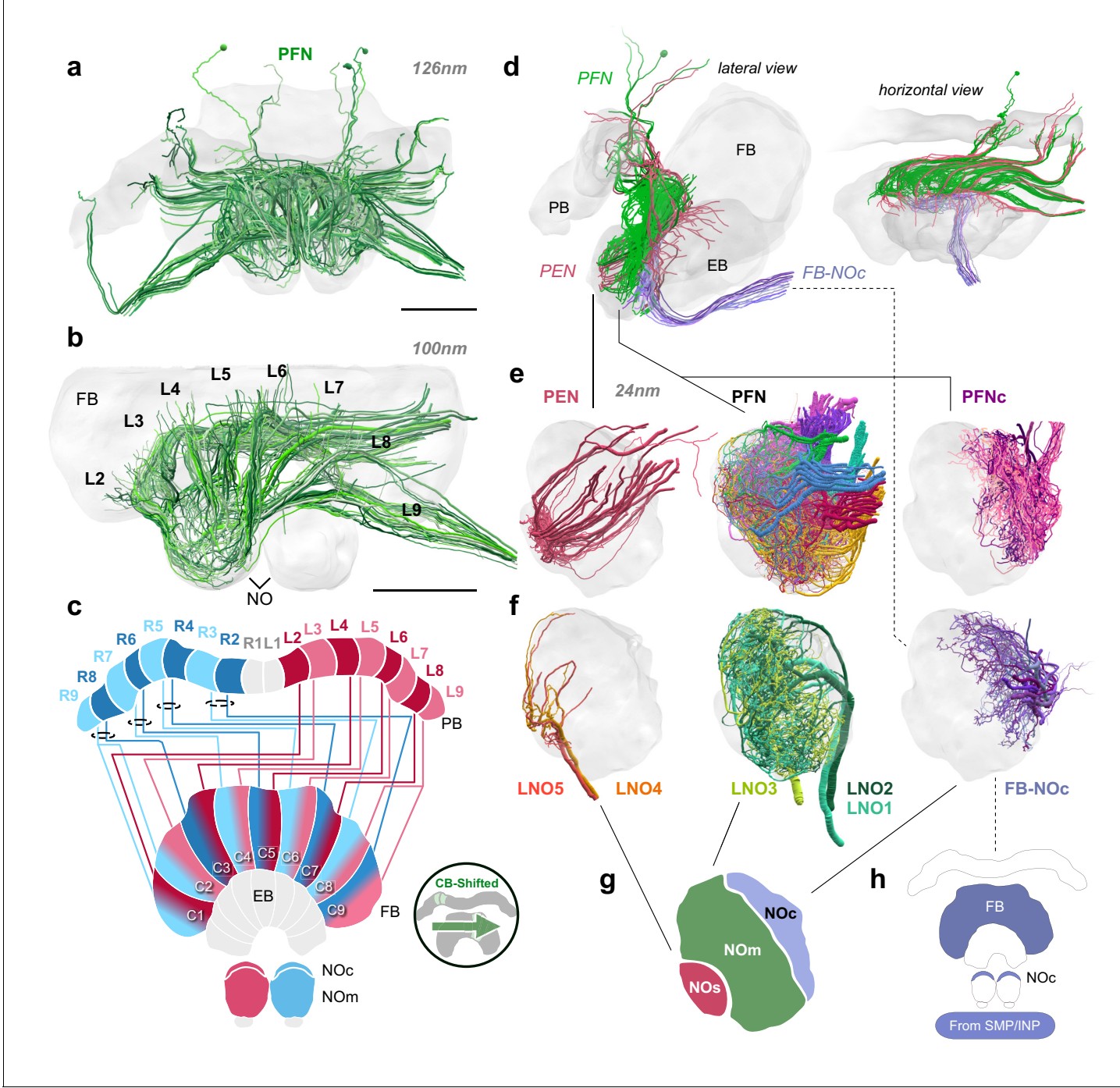

**Figure 4.** Structure and cellular composition of the NO. (**a**) Frontal view of PFN cells from the 126 nm data set. (**b**) Frontal view of PFN cells in 100 nm data set projecting from PB columns L2-L9. (**c**) Schematic of PFN cell projection patterns. PFNs are shifted contralaterally by one column in the FB. (**d**) Lateral and horizontal views of PFN (green), PEN (pastel red), and FB-NOc (purple) cells. (**e–f**) Reconstructions from 24 nm data set. (**e**) Lateral view of PEN, PFN, and PFNc innervations in the NO, each group occupying a discrete zone. (**f**) NO tangential cells (LNO1-5) and FB tangential cell (FB-NOc) fibers reveal at least three structural compartments schematized in (**g**). To avoid confusion with the fruit fly NO zones, which are named by anatomical position (*Wolff and Rubin, 2018*), we have called these the small unit (NOs), main unit (NOm), and the cap (NOc). (**h**) Colored regions show the arborization domains of FB-NOc cells. Scale bars: (**a–b**) 100 μm.

The online version of this article includes the following figure supplement(s) for figure 4:

**Figure supplement 1.** PFN bundles innervating the NO and FB-NOc cell morphology.

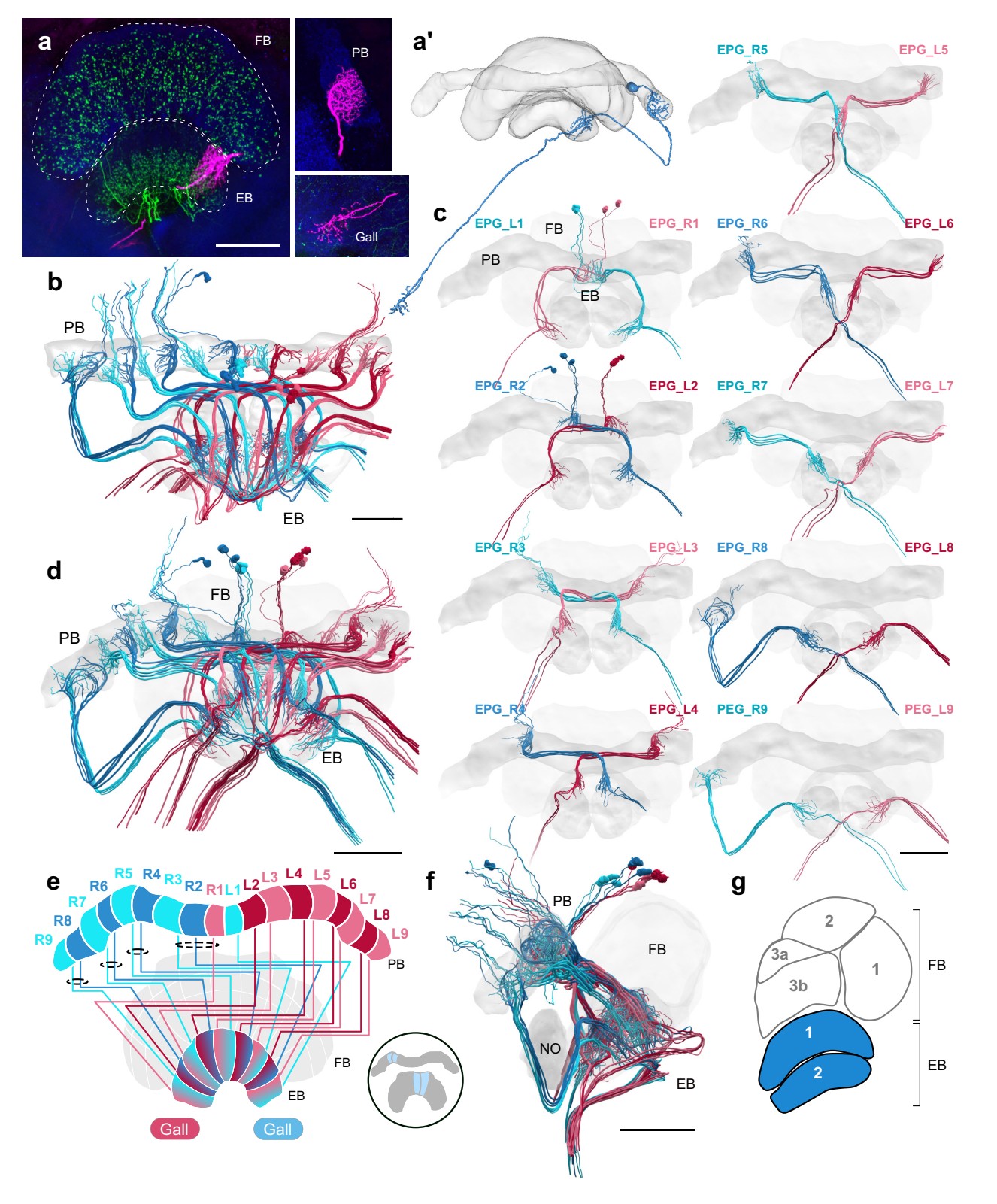

**Figure 5.** EPG/PEG cells. (a) Intracellular dye injection of a bumblebee EPG/PEG cell (magenta), counterstained with serotonin (green) and synapsin (blue). (a') Reconstruction of cell in (a). (b–d,f) Neural reconstructions of EPG/PEG cells from the 126 nm data set. (b) Horizontal view of all EPG/PEG cells. (c) Frontal view of isolated EPG/PEG cells projecting from each PB column (labeled 'EPG' here for simplicity). 'L-' and 'R-' refer to the associated PB column for each cell. Blue neurons project from right hemisphere and red from left. (d) Frontal view of all EPG/PEG cells. (e) Schematic of EPG/PEG

*Figure 5 continued on next page*

Figure 5 continued

projection pattern. EPG/PEG cells follow the 'default' projection pattern in the EB. (f) Lateral view of all EPG/PEG cells. (g) Lateral schematic with EPG/
PEG innervating layers in blue. Scale bars: (a) 50 µm; (b–d,f) 100 µm.

minimum of 160 PFN neurons in a single hemisphere, with many still left to be traced. Finally, with the 24 nm data set, and including the PFNc neurons, we were able to identify a total of 427 PFN neurons in one hemisphere (*Figure 3—figure supplement 1*), with a maximum number of 100 PFN cells projecting from a single column in the PB/CB (*Figure 4—figure supplement 1d*; R4). Based on these numbers, we estimate the total numbers of PFN cells to be at least 854 for the entire CX, double the number found in *Drosophila* (*Hulse et al., 2020*). Interestingly, approximately half of the found PFN cells belong to the PFNc subtype (201 in the analyzed hemisphere).

At least two thirds of PFN neurons had very small fiber diameters, making them impossible to trace using the 126 nm resolution data set. Additionally, as branches within the FB could only be traced over short distances and no layering was present in the NO, we were unable to subtype this large group of neurons further, leaving PFNc and PFN (main) as the only obvious subtypes. However, clear and consistent differences in the diameter of the main PFN neurites in each columnar bundle demonstrated that the population of PFN cells is not homogeneous and many subtypes likely exist. With regard to their columnar projection pattern, PFN cells are laterally shifted by one column in the FB and are entirely absent from the innermost PB columns (*Figure 4c*). In this regard, they are identical to the PEN cells that connect the NOs to the EB (see next section; Figure 6d).

## A fly-like head direction circuit in the bee ellipsoid body

In the bumblebee EB we found two principle types of columnar cells, EPG/PEG cells (*Figure 5*) and PEN cells (*Figure 6a–c*). While in other species, the former can be further classified into two subtypes based on their opposite polarity, this distinction could not be made in our data, owing to the limited resolution of the data set. EPG/PEG cells as well as PEN cells both carry information between the PB and EB. While PEN neurons additionally arborized in the NOs, where their fibers overlapped with tangential LNO4/LNO5 cells (*Figure 6c*), EPG/PEG cells extended an axon into a small region positioned laterally and anteriorly to the CX, close to the border of the protocerebrum and the antennal lobe (*Figure 5a–a'*). Based on the morphology of EPG/PEG neurons in other species, we concluded that this target region corresponds to the gall, despite the lack of obvious neuropil boundaries in synapsin labeled preparations.

In the fruit fly, PEN and EPG/PEG cells (hereon referred to as EPGs for simplicity) form the core of the heading direction circuit (*Turner-Evans et al., 2020*). Akin to a biological compass, EPGs track the flies rotational movement as a single bump of activity in the EB and two bumps of activity in the PB, one in each hemisphere (*Seelig and Jayaraman, 2015*). Tangential EB neurons feed predominantly visual information from the anterior visual pathway to EPGs in the EB, tethering the bump to features of the environment (*Fisher et al., 2019*; *Kim et al., 2019*). At the same time, rotational self motion cues are sent to the PEN cells, predominantly via the NO and probably also directly via the PB (*Turner-Evans et al., 2017*; *Green et al., 2017*). While activity in the EB input cells directly generates an activity bump in the EPG network that is based on allothetic information, idiothetic information laterally shifts the bump position via the PEN activity. Importantly, this circuit relies on an anatomical offset between the EB projections of PEN and EPG cells: EPGs follow a default projection pattern, whereas PENs are shifted by one column in the EB. Therefore, EPGs projecting to any given column in the EB will be flanked on either side by PEN fibers, connecting to neighboring EPG cells. This offset has the effect whereby PEN activation results in shifting the activity bump either to the left or to the right, thereby translating clockwise or counterclockwise body rotations into counterclockwise or clockwise movements of the neural activity in the EPG population (*Turner-Evans et al., 2017*; *Green et al., 2017*).

An identical offset in projections between EPG/PEG and PEN cells was also present in the bumblebee CX: EPG/PEG cells followed the default projection pattern (*Figure 5e*), whereas PEN cells were shifted by one column in the CB (*Figure 6d–f*; *Figure 6—figure supplement 1*). Moreover, in the bee the total number of EPG/PEG and PEN cells per CX column closely approximated their homologues in the fly. In total, 69 EPG/PEG cells spanned the width of the bee CX with four neurons

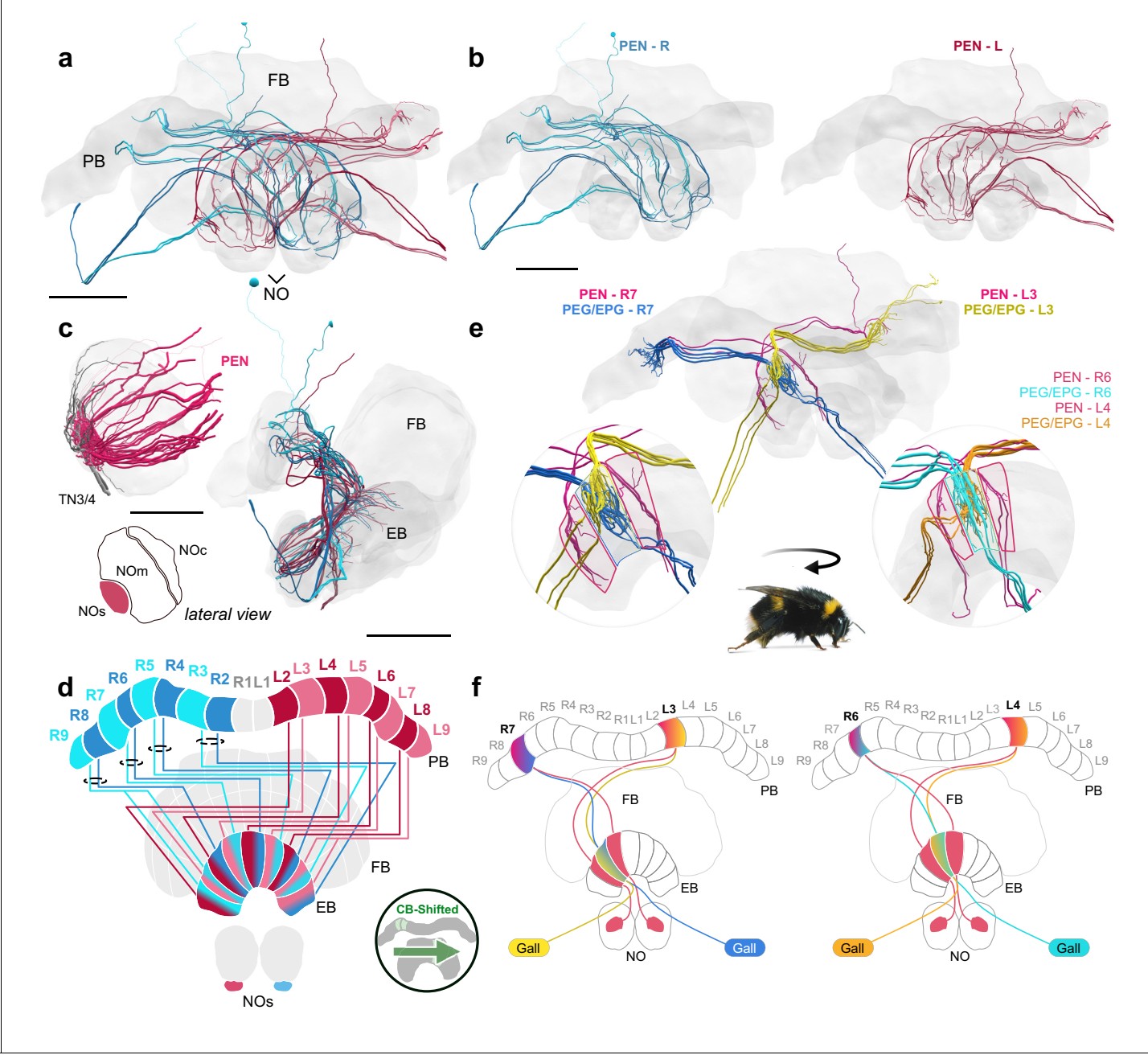

**Figure 6.** A fly-like head direction circuit in the bumblebee EB. (**a–b**) Frontal views of PEN 'angular velocity' neurons projecting from left (red) and right (blue) hemispheres. (**c**) Lateral view of PEN arborizations in the NOs in 24 nm and 126 nm data sets. (**d**) Schematic of PEN projection patterns. PEN cells are shifted contralaterally by one column in the EB. (**e**) An example illustrating how the anatomical projections of EPG/PEGs and PENs could support the shifting of an EPG activity bump as the bee rotates. (**f**) Schematic illustration of (**e**). Scale bars: (**a–b**) 100 μm (**c**) 25 μm 24 nm data set; 100 μm 126 nm data set.

The online version of this article includes the following figure supplement(s) for figure 6:

**Figure supplement 1.** EPG-PEN projectivity across EB columns.

present in each PB column except for the innermost columns, which only contained three neurons each (*Figure 5b–d*; *Figure 3—figure supplement 1*). In the 126 nm data set, we identified 34 PEN cells at two per PB column, with the exception of the outermost columns that contained three PEN cells and the innermost columns that were entirely devoid of PEN neurons (*Figure 6a–b,d*; *Figure 3—figure supplement 1*). Interestingly however, we found 20 PEN cells projecting from a single

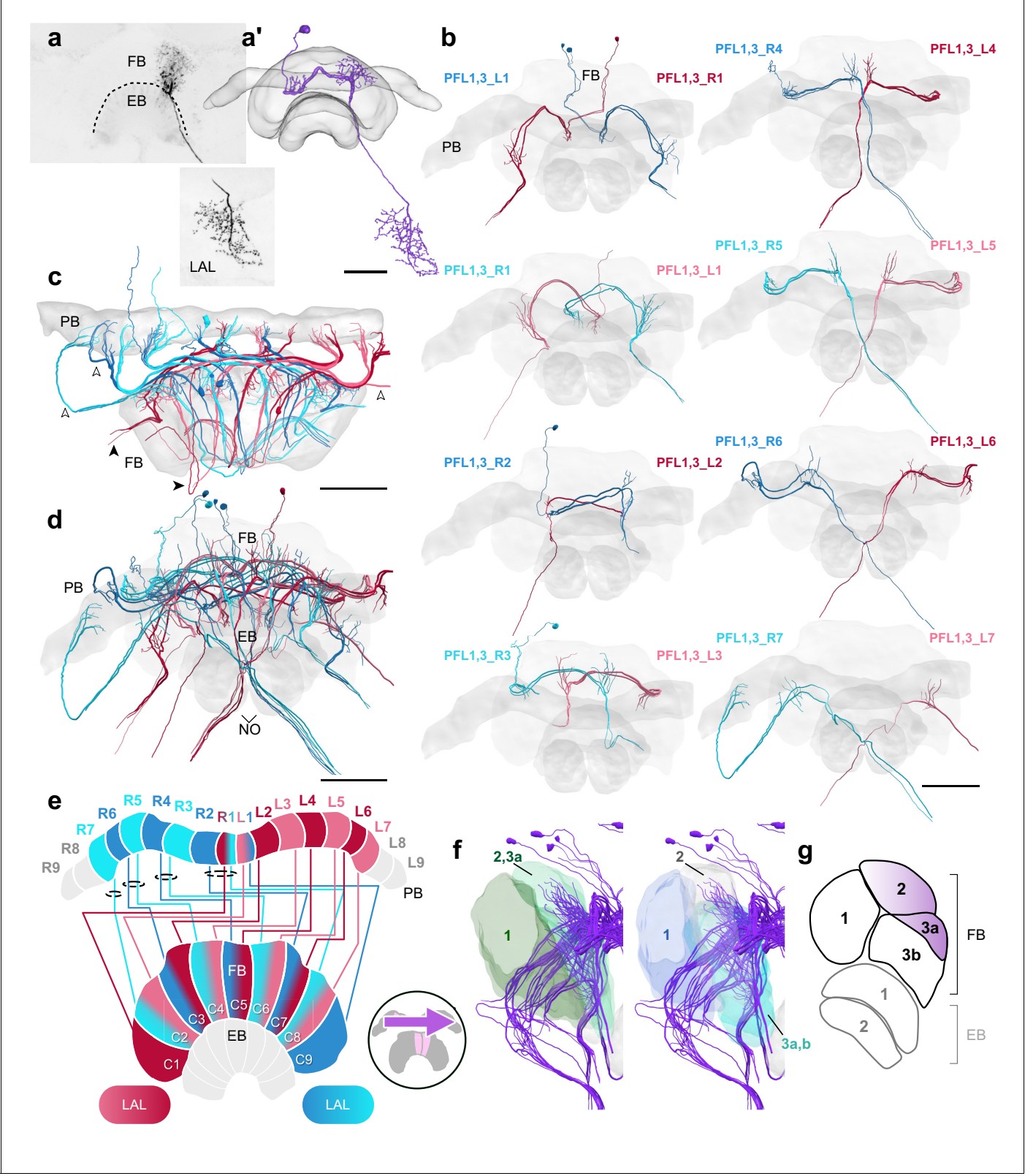

**Figure 7.** PFL1,3 cells. (**a**) Intracellular dye injection of a *Megalopta genalis* PFL1,3 cell. (**a′**) Reconstruction of cell in (**a**). (**b–d,f**) PFL1,3 cells traced from 126 nm data set. (**b**) Frontal view of isolated PFL1,3 cells projecting from each PB column. 'L-' and 'R-' refer to the associated PB column for each cell. Blue neurons project from right hemisphere and red from left. (**c**) Horizontal view of all PFL1,3 cells. PFL1,3 neurons are characterized by large diameter fibers leaving the PB (white arrowheads) which thin substantially as they exit the FB (black arrowheads). (**d**) Frontal view. (**e**) Schematic of PFL1,3

*Figure 7 continued on next page*

*Figure 7 continued*

projection patterns. PFL1,3 cells follow the 'PB-shifted' projection pattern. They are ipsilaterally offset by a single column in the FB. (**f**) Lateral view of PFL1,3 cells relative to layers defined by serotonin (green) and tyrosine hydroxylase (blue) immunolabeling. (**g**) Lateral schematic with purple gradient approximating the innervation regions of PFL1,3 cells. Scale bars: (**a**) 50 µm; (**b–d**) 100 µm.

hemisphere in the 24 nm data set, which suggest a total of 40 PEN cells are present in the entire bee CX, nearly identical to numbers to those found in *Drosophila* (*Hulse et al., 2020*). In the fly, 74 EPG/PEGs and 42 PENs span the width of the CX with a similar, but slightly different distribution of these cells per PB column (See *Hulse et al., 2020*).

Notably, we found one set of EPG/PEG cells with arborizations in the innermost PB columns of the contralateral hemisphere ('EPG_L1' and 'EPG_R1' in *Figure 5c*). In contrast to the rest of the EPG/PEGs, these cells contained branching fibers in the outermost EB columns of the ipsilateral hemisphere relative to their arbor in the PB (*Figure 6—figure supplement 1*). These cells may provide a pathway for an activity bump to 'jump' from one lateral end to the other in the bumblebee EB, which compared to *Drosophila* is structurally an open loop (see Discussion).

## The fan-shaped body

The FB is the largest and arguably the most functionally mysterious of the CX neuropils. In our data set, the majority of FB columnar cell types were found to connect the PB with the FB and onward to tertiary regions. They possessed arborizations in the FB where they commingle among themselves, FB tangential neurons, and interneurons that were confined solely within the FB, the hΔ cells, or pontine cells (Figure 11; e.g. *Hanesch et al., 1989*).

As with the EB, we focused our reconstructions on the columnar cells with large neurite diameters. We revealed at least nine types of FB columnar cells in the bumblebee CX (*Figure 3b*). One group of cells, the already mentioned PFN cells, innervated the posterior portion of the FB and represented the most numerous columnar neurons of the FB. Even considering only the largest PFN cells obtained in the low resolution data (202 individuals) their number exceeded the total count of all remaining columnar FB cells combined (163 neurons). This ratio is likely even more biased toward PFN cells when considering that the 202 PFN cells likely comprise only one fourth of the total number of these cells (as identified in our high-resolution data set). As the extent of the fine processes within the FB was not traceable, the exact layer these cells project to could not be identified. Although fiber diameters of different sizes present in each columnar bundle of PFN cells demonstrates the existence of subtypes, these could not be reliably established.

The limits imposed by the unresolved neuronal terminals apply to all neuron types of the FB and were amplified by the fact that neural processes outside the imaged tissue block were truncated. Nevertheless, all identified cell types were clearly distinct from each other based on their columnar projection patterns (*Figure 2e–f*), the fiber trajectories within the FB (*Figure 2b'*), the exit point they left the FB, and the location of the main branches.

Two types of neurons had main neurites located near the dorsal edge of the bundle connecting the PB with the FB and possessed characteristically large fiber diameters when leaving the PB (*Figure 7*; *Figure 8*). This main neurite entered the FB dorsally giving rise to numerous thin branches. While passing through the FB in a ventral direction this neurite dramatically, but consistently, thinned before turning toward the contralateral lateral complex (*Figure 7f*; *Figure 8e*). Due to this thinning, many cells were lost at that point. All these cells followed a projection pattern that exhibited shifted innervations of the PB. Based on these features as well as a single dye filled example from the halictid sweat bee *Megalopta genalis* these cells were identified as PFL neurons, the main output cells of the CX. More detailed comparisons of the projection patterns with data from other insects (*Heinze et al., 2013*; *Hensgen et al., 2021*; *Hulse et al., 2020*) allowed defining them as PFL1/3 as well as PFL2 neurons. While the latter has a bifurcating neurite in all other species, the bifurcation was not identified in our data, very likely because of the extremely thin neurite diameter at the point where the bifurcation should be located.

PFL2 neurons were unique in other aspects as well. Although not shown in the idealized schematics (*Figure 8d*), these cells did not typically stick within well defined columns in the PB. For instance, PFL2 fibers that arborized in the innermost PB columns showed considerable overlap, and

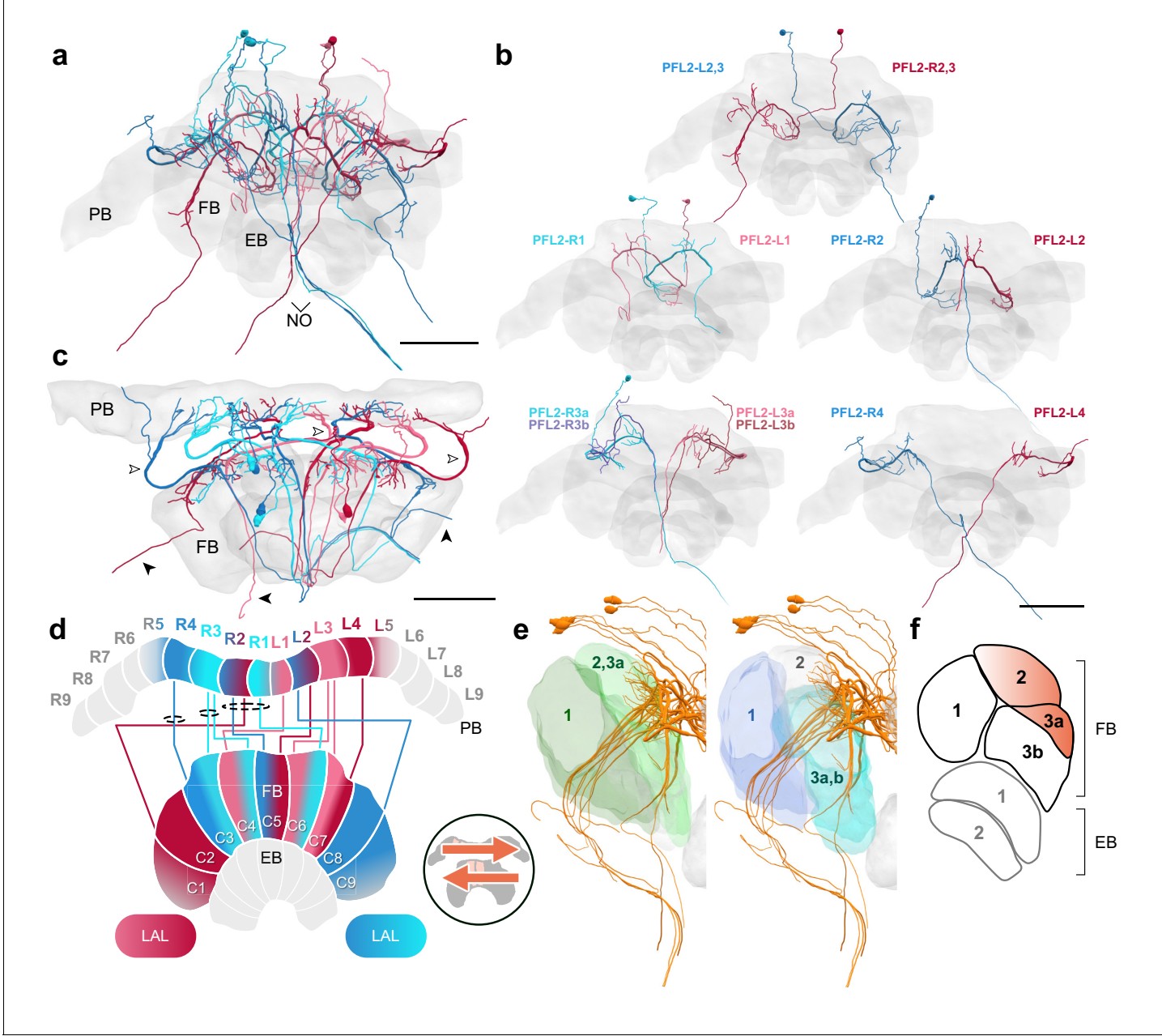

**Figure 8.** PFL2 cells. (a–c,e) PFL2 cells traced from 126 nm data set. (a) Frontal view of all PFL2 cells. (b) Frontal view of isolated PFL2 cells projecting from each PB column. 'L-' and 'R-' refer to the associated PB column for each cell. Blue neurons project from right hemisphere and red from left. (c) Horizontal view of all PFL2 cells. Like PFL1,3, PFL2 neurons are characterized by large diameter fibers leaving the PB (white arrowheads) which thin substantially as they exit the FB (black arrowheads). (d) Schematic of PFL2 projection patterns. Note from (c) that, unlike most columnar cells, PFL2s have substantially overlapping fibers in the PB. Therefore, this schematic is simplified. Columns in the PB were determined by the location of EPG fibers (*Figure 5*). PFL2 cells follow the 'both-shifted' projection pattern. They are ipsilaterally offset by three columns in the FB, with the exception of a single neuron in both hemispheres (R3 and L3) that is shifted ipsilaterally by four columns (PFL2_R3b and PFL2_L3b). (e) Lateral view of PFL2 cells relative to layers defined by serotonin (green) and tyrosine hydroxylase (blue) immunolabeling. (f) Lateral schematic with orange gradient approximating the innervation regions of PFL2 cells. Scale bars: 100 μm.

in some cases PFL2 fibers were found covering a span of around three columns in the PB, relative to the projection domains of EPG/PEG cells (see for example *Figure 14—figure supplement 1*).

Two more groups of columnar FB cells were identified, all of which followed the default projection pattern, identical to that shown by EPG/PEG cells of the EB. The first set of cells was a second

type of columnar cell intrinsic to the CX, as no fibers were found that exited the CX after branching in the FB (*Figure 9a–d*). These cells were termed PF1 cells and existed in two copies per PB column. Whereas fibers within the PB were not resolvable in each individual, in some columns these fibers were sufficiently clear to suggest their existence across all individuals of this set. These neurons were the only cells innervating predominantly the anterior and ventral layers of the FB with numerous processes (*Figure 9f*). Intriguingly, some fibers appeared to pass the border between the FB and the EB, sending fine processes into the EB. While this needs to be confirmed by higher resolution data, PF1 cells might be uniquely innervating both the FB and the EB (*Figure 9d,f–g*).

Finally, a set of four related cell types was found in more ventral regions of the W, X, Y, Z bundles and traversed the FB in a shared fiber tract (*Figure 10*). Their main neurite was comparably thin when entering the FB dorsally, but substantially thickened when giving rise to projections inside the FB, making these cells clearly distinct from PFL neurons. As their main neurite was very thin when entering the PB, it could rarely be traced inside the PB, yielding only a few examples with verifiable branches within the PB (*Figure 10*). This leaves some doubt as to whether all these cell types indeed posses significant PB arborizations. For two of the four cell types, PB branches were confirmed by intracellular dye fills (*Figure 10*; *Figure 10—figure supplement 1*). Assuming that PB branches are a feature of all these cells, the four types were named PFx1-4 (*Figure 10*).

Two of the four cell types (PFx3,4) sent bilaterally reaching fibers toward the superior medial protocerebrum (SMP) and/or the anteriorly located portions of the inferior neuropils (INP; *Figure 10*). Similarly, but without bifurcation, PFx1 neurons leave the FB on the anterior side, but turn toward the contralateral anterior brain, wrapping around the anterior surface of the medial lobe of the mushroom body (*Figure 10*). The target of these cells was unclear and might either be the lateral complex, or the anterior regions of the inferior protocerebrum. In contrast, PFx2 cells clearly sent fibers toward the lateral complex and ventromedial protocerebrum (VMP) along a similar trajectory as PFL cells and EPG/PEG cells, with their axon passing the medial lobe of the mushroom body posteriorly.

An additional, unusual type of cell included neurons with large diameter fibers that occurred in two individuals in one hemisphere and four in the other. These cells resembled PFL neurons in that they possessed very large neurites near the PB, which then dramatically thinned after passing the FB and turns toward the contralateral lateral complex. These neurons did not give rise to resolvable branches within the FB, but the existence of fibers below our resolution limit cannot be ruled out. Due to their resemblance with PFL cells, they are also assigned to the FB and were named PFLx neurons (*Figure 3b*). Interestingly, this cell type shows a clear asymmetry in that one individual in column L1 (projecting to the ipsilateral lateral complex) does not have a counterpart on the contralateral side.

## Fan-shaped body interneurons: hΔ cells

The FB is the only CX neuropil which generally contains a system of interneurons that are confined solely within its boundaries. These neurons contain input fibers in one column and output fibers in a column in the opposite hemisphere (*Heinze and Homberg, 2008*; *Hulse et al., 2020*). Historically, these cells have been termed pontine cells (*Hanesch et al., 1989*), but have been renamed as hΔ neurons in the fly literature given their proposed role as neurons that horizontally shift (i.e. horizontal-shift: hΔ) neural activity from one column of the FB to another column in the opposite hemisphere. Here, we adhere to the new nomenclature proposed by *Hulse et al., 2020* and refer to these cells as hΔ (*Figure 11*).

Similar to columnar cells, hΔ neurons possessed cell bodies residing in the dorsal-posterior regions surrounding the CX (*Figure 11*). We identified as many as 188 hΔ neurons that tile the FB into vertical columns, very similar to the 190 hΔ neurons identified in the fly by *Hulse et al., 2020* (*Figure 11*). This high number makes them the second most numerous cell type of the CX after the columnar PFN cells. Their numbers were surprisingly unevenly distributed across the bundles of origin. Interestingly, the X and Y bundles on both hemispheres contained almost exactly twice the number of hΔ cells compared to the W and Z bundles (*Figure 11*). This suggested an overall branching system consisting of twelve lateral sectors across the width of the FB. Indeed, the cells originating in the X and Y bundles could be segregated into two adjacent arborization domains each (*Figure 11*).

Most hΔ neurons possessed a midline crossing neurite passing the FB posteriorly (purple in *Figure 11*), while other, rarer types passed the midline along a more dorsal (orange) or anterior course

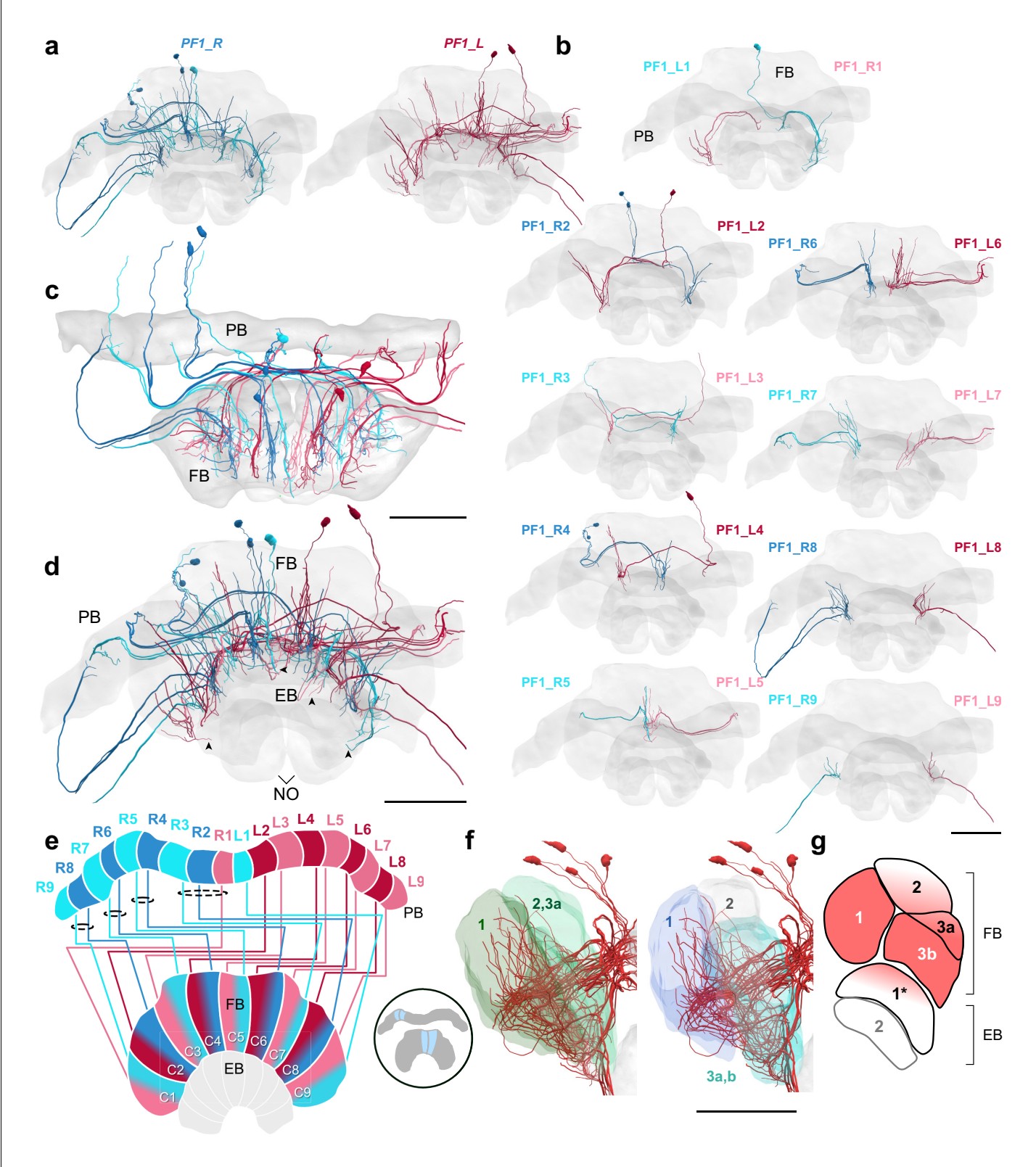

**Figure 9.** PF1 cells. (**a–d,f**) PF1 cells traced from the 126 nm data set. (**a**) Frontal view of all right hemisphere PF1 cells (blue) and left hemisphere PF1 cells (red). (**b**) Frontal view of isolated PF1 cells projecting from each PB column. 'L-' and 'R-' refer to the associated PB column for each cell. (**c**) Horizontal view. (**d**) Frontal view. Arrowheads indicate PF1 fibers that enter into the EB from the FB. (**e**) Schematic of PF1 projection patterns. PF1s follow the 'default' pattern in the FB. (**f**) Lateral view of PF1 cells relative to layers defined by serotonin (green) and tyrosine hydroxylase (blue)

*Figure 9 continued on next page*

*Figure 9 continued*

immunolabeling. (**g**) Lateral schematic with red gradient approximating the innervation regions of PF1 cells. Note that some cells have ventral branches that extend into the EB (asterisk). Scale bars: 100 μm.

(green; *Figure 11*). The collective arborizations of the majority of these posterior cells followed the described twelve vertical column system across the FB (*Figure 11*). However, most cells that belonged to the dorsally (*Figure 11*) and anteriorly projecting hΔ groups (*Figure 11*), as well as one posterior hΔ cell type (Type 4) appear to form only eight vertical columns. These cells shared projections within the anteriormost layer of the FB.

Based on their morphology, relative distribution, and their arborization layer within the FB, we identified a minimum of five hΔ types (*Figure 11*). hΔ Type 1 contained minute fibers that innervated only the posterior surface of the FB and arborized in the dorsalmost layers. hΔ Type 2 cells had slightly longer projections within the FB, but arborized in the ventralmost layers. hΔ Type 3 cells occupied layers between hΔ cell Types 1 and 2, and had longer FB projections than both types. hΔ Type 4 cells had branches that spanned the entire posterior-anterior axis of the FB, and appeared to occupy the same horizontal domain as Type 3 cells. The branching fibers which innervated the FB furthest away from the location of their cell bodies, that is, their likely output fibers, were positioned anteriorly relative to the domains of their putative input fibers. Further, these cells had wider branches and collectively formed eight columns as opposed to the twelve formed by Types 1, 2, and 3. hΔ Type 5 cells had wide-branching fibers in the dorsal regions of the FB. Interestingly, the putative input fibers of these neurons entered the FB posteriorly, but their putative outputs enter dorsally and are reminiscent in morphology of a claw crane (*Figure 11*). Lastly, all anterior hΔ cells were combined into Type 6 (*Figure 11*).

Due to the limited resolution of our data set, we were unable to trace finer branches and could not determine the FB layer in which these cells send their terminal branches. Our analysis is therefore only a first approximation and it is highly likely that more types of hΔ cells exist.

## Discussion

In this first study of a hymenopteran CX based on 3D EM data, we asked what neuroarchitectural information would be attainable by tracing a relatively low resolution SBEM data set of the entire CX of a *Bombus terrestris* worker. To this aim we manually traced more than 1300 columnar and pontine neurons of the bumblebee CX and established neural projection patterns underlying overall information flow within this brain region. Based on these patterns, we extrapolated whether anatomically constrained computational algorithms identified in particular in the fruit fly could also exist in the bumblebee. The present study is therefore a first step toward evaluating how conserved the neural circuits uncovered by the *Drosophila* connectome (*Scheffer et al., 2020*; *Hulse et al., 2020*) are and which of these circuits are general features of insect brains and which ones are specific adaptations relevant to fruit fly ecology and behavior.

### What information is missing from our low-resolution projectome?

Additionally to the low-resolution data, we collected higher resolution data sets of the NO from a different individual. This not only allowed us to trace the fine neurites of PFN cells, which have been proposed to integrate heading with distance information in bees and likely play a major role in path integration (*Stone et al., 2017*), but also enabled us to estimate how many neurons we are likely missing when relying on low resolution data only.

The comparison between the three data sets revealed that the lowest resolution data clearly misses all neurons with neurite diameters smaller than a certain cutoff, in our case 0.6 μm. This cutoff was not identical across all regions of the data, as especially toward the boundaries of the main image stack, resolution was much poorer compared to the center.

Additionally, the inability to resolve fine details also led to the loss of all but the main branches of arborizations within neuropils. For instance, in the noduli, the PFN neurons as well as the LNO input cells possessed only fibers totalling a few μm in the 126 nm data, but possessed cable lengths of several orders of magnitude larger when reconstructed based on the higher resolution data. Synapses were not clearly visible in any of the three data sets.

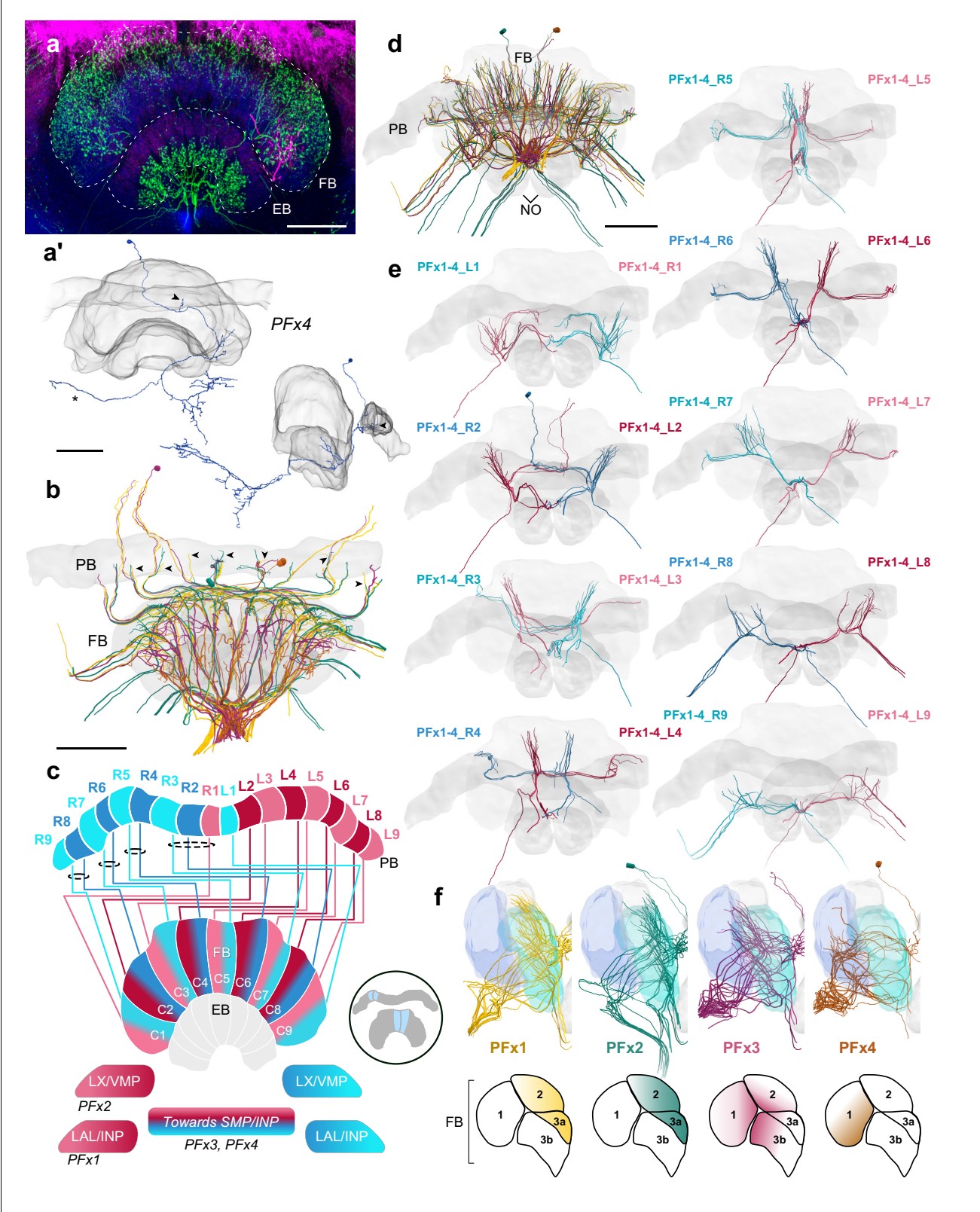

**Figure 10.** PFx cells. (a) Intracellular dye injection of a bumblebee PFx4 cell (magenta) co-stained with serotonin (green) and synapsin (blue). (a')
Reconstruction of cell in A. Arrowheads point to fibers entering the PB. Reconstruction is partially incomplete due to weak signal in overview confocal
scan (asterisk), see ***Figure 10—figure supplement 1d*** for additional high resolution image data. (b,d–e,f) All four types of PFx cells traced from 126 nm
data set. (b) Horizontal view of all PFx types, illustrating the differentially innervated FB layers. Arrowheads point to fibers entering the PB. (c) Schematic

*Figure 10 continued on next page*

*Figure 10 continued*

of PFx cell projection patterns. PFx1 cells send their fibers away from the CX toward regions in the INP/LAL, while PFx2 contains fibers ventrally descending toward the LX. PFx3,4 cells project toward regions in the SMP/INP. All PFx cells follow the 'default' pattern in the FB. (**d**) Frontal view. (**e**) Frontal view of PFx1,2,3,4 cells projecting from each PB column. 'L-' and 'R-' refer to the associated PB column for each cell. Blue neurons project from right hemisphere and red from left. (**f**) Lateral view of PFx cells relative to layers defined by tyrosine hydroxylase (blue) immunolabeling. Lateral schematics underneath show approximate innervation regions of PFx1,2,3,4 cells within the FB. Scale bars: 100 μm.

The online version of this article includes the following figure supplement(s) for figure 10:

**Figure supplement 1.** Intracellular dye injection of PFx2 and PFx4 cells.

This not only has implications for detailed maps of overlapping arborizations, but limits cell type identification. For instance, E-PG/P-EG cells both occupy similar regions in the PB, EB, and gall, share projection patterns and morphology, but differ in polarity, with one cell type receiving input predominantly in the EB (E-PG) and the other in the PB (P-EG). The information needed to resolve these cell types clearly lies in higher resolution data. Other examples are PFL1 versus PFL3 cells, as well as subtypes of PFN or hΔ cells, all of which, in *Drosophila*, are defined based on the innervated layer of the FB and, ultimately, by their underlying connectivity. Without higher resolution and the associated ability to reconstruct full branching trees, we are unable to discern the subtype identity of any of these cells in the bumblebee.

Adding a different perspective, we also compared the skeletons of our low-resolution EM-based data to detailed neuron morphologies obtained by intracellular dye injections and confocal microscopy. This analysis not only demonstrated that regions of cells that lie outside of the imaged volume are obviously missing from our data, but that our low-resolution EM data can be supplemented by full arborization trees obtained by light microscopical data, as long as neurons are selectively labeled. The complete morphologies obtained with this method confirmed the result of the comparison between low- and high-resolution EM data and highlighted that neither polarity indicators (fine versus beaded endings), nor the full size of branches were captured in the EM-based reconstructions based on low-resolution images.

However, mapping of EM-based skeletons and neuron skeletons based on confocal data yielded surprisingly good matches, even in the occasional cases when data from *M. genalis* was mapped onto bumblebee data. This indicated that the gross morphology of neurons can be readily compared across these imaging modalities and future data based on single neuron dye fills can indeed be used to validate identity of neuron types in our projectome data.

We conclude that, despite all the missing details, known cell types can be reliably identified based on our data. Sub-classifications of these main types that are based on either fiber polarity or details of innervated layers are, however, not possible and require additional data for cross validation. Mostly owing to the missing projections outside of the CX, the morphology of unknown types of neurons are also impossible to determine without additional verification based on dye-filled preparations. Nevertheless, by using the presented data, even an individual example of a dye filled neuron is sufficient to assign identity to all individuals of this type in our projectome. Each complete neuron morphology therefore automatically provides information about the entire isomorphic set of its kind. The existence of our projectome thus substantially increases the value of single neuron morphologies obtained in future studies.

## Novel insights into the overall layout of the insect CX

For decades CX research did not question that the CX contains 16 columns in the PB (e.g. *Hanesch et al., 1989*; *Heinze and Homberg, 2008*; *El Jundi et al., 2018*). Based on work in the locust, this assumption was used to draw functional conclusions for example about the basis of representing compass directions (*Heinze and Homberg, 2007*). More importantly, it was implied by generalizations across species, that the CX organization in one species (e.g. the locust) was equivalent to that of others. This was well justified by the developmental origin of the columnar CX neurons, which, across insects, are derived from eight neuroblasts, four on either side of the midline (*Boyan and Williams, 2011*). Each of these gives rise to one of the W,X,Y,Z bundles and thereby provides neurons for two PB columns (*Williams et al., 2005*; *Williams and Boyan, 2008*).

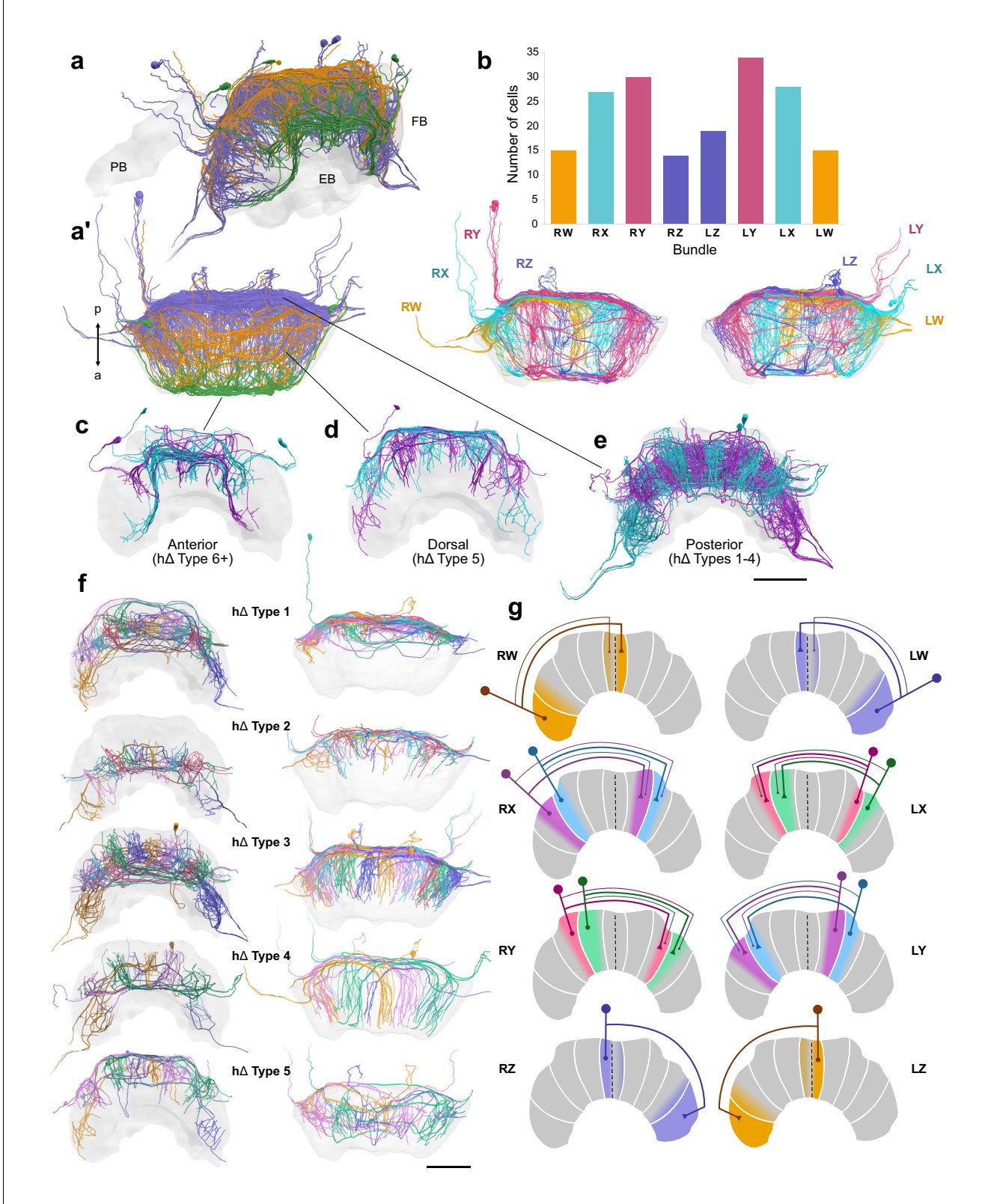

**Figure 11.** hΔ cells. (a–f) Reconstructions of hΔ cells from 126 nm data set. (a) Oblique and (a') horizontal views of hΔ cells entering the FB posteriorly (purple), dorsally (orange), and anteriorly (green). (b) Numbers of hΔ cells per CX bundle. Below, horizontal view of hΔ cells in each hemisphere colored according to bundle. (c–g) Cells are colored according to the FB column they arborize in. (c) Anteriorly projecting hΔ cells. These include Type 6 hΔ cells. Plus sign denotes the possibility that more subtypes of anteriorly projecting hΔ cells are likely to exist. (d) Dorsally projecting hΔ cells. (e)

*Figure 11 continued on next page*

*Figure 11 continued*

Posteriorly projecting hΔ cells. (f) There are at least five types of hΔ cells which clearly differ in their projection layer and morphology. More cell types may exist but would require full cell morphologies from high-resolution tracing data. Frontal views (left) and horizontal views (right). (g) Schematics of hΔ FB projection columns according to bundle. hΔ cells send putative input fibers to the column nearest their soma, and putative output fibers to FB columns shifted 3–4 columns contralateral to their inputs (see also *Figure 11—figure supplement 1a*). Scale bars: 100 μm.

The online version of this article includes the following figure supplement(s) for figure 11:

**Figure supplement 1.** FB hΔ cells isolated by bundle and viewed laterally by type.

However, detailed analysis using genetic methods in the fruit fly revealed that the fly PB consists of nine instead of eight columns (glomeruli; *Wolff et al., 2015*). This significantly challenged the notion that data between species could be easily generalized and raised the question of how to homologize neurons across species.

Our data clearly revealed that, like the fly (*Wolff et al., 2015*), the bumblebee PB also comprises nine columns in each PB hemisphere (*Figure 2c–d*). As in most other insects, vertical columns are not visible using immunohistochemical labeling with antibodies against synapsin, neither in the FB, EB or PB, and a definite insight into this structural feature required EM based tracing of main fiber trajectories of the CX.

Interestingly, the existence of nine columns in the EB was already reported by *Williams, 1975* and was confirmed also for bees, e.g. by *Hensgen et al., 2021*. In each case, seven wide, medial columns are flanked by two narrow hemicolumns at the lateral edges of the EB, generating eight equal sized projection fields along the length of the EB. The division of the fly EB into eight equally wide segments (referred to as 'tiles' *Wolff et al., 2015*) around the EB ring is directly equivalent and suggests that the existence of eight versus nine columns in the PB does not affect the overall structure of the CX (*Figure 12*). Rather, the ninth column likely evolved after the core circuitry of the CX.

This hypothesis is consistent with the eightfold symmetry of the CX head direction circuit in flies (*Hulse et al., 2020*; *Pisokas et al., 2020*). It is also supported by our data, which show that most divergence in projection patterns between the fly and the bumblebee neurons was found at the innermost PB column (*Figure 2c*), that is this additional column provides the substrate to evolve novel CX components without disrupting the existing functions based on the outer eight columns. This also suggests that the columns 2–9 in flies and bees are homologous to columns 1–8 in locusts and other insects. The consistent lack of PFN and PEN neurons in the innermost columns further supports this notion (*Figure 4c*; *Figure 6d*).

Underlining the high degrees of organizational conservation, we were able to identify four discrete columnar cell projection patterns (*Figure 2e–f*), three of which were identical to the patterns produced by the corresponding neurons in the fly. Importantly, these projection patterns define the computations that are carried out within the EB and the FB. This resemblance in structure therefore suggests that highly conserved computational principles operate at the core of the CX across insects. For instance, functional recordings and connectomics analysis in the fly have led to a model in which navigation-related vector computations are carried out by columnar cells and interneurons in the FB (*Hulse et al., 2020*; *Lu et al., 2020*; *Lyu et al., 2020*). Columnar cell projection patterns provide the structural backbone of these computations and would therefore define which vector operations can be achieved. Using the same argument, those patterns that were identified as being different, namely those of PFL2 neurons in bees and of PFL3 neurons in flies, directly suggest that distinct computations are implemented in either species, a finding that will be discussed in more detail below.

On the level of neuropils, and in contrast to the strictly conserved columnar organization, the layers of the CB identified via immunohistochemical staining against serotonin (5HT) and tyrosine hydroxylase (TH, an enzyme required to produce dopamine), bear no close resemblance between bees and flies. We found at least three horizontal layers in the FB and two layers in the EB (*Figure 1d–e''*), while, based on immunolabeling of synaptic markers alone, many more layers are present in flies (*Wolff et al., 2015*), as well as in other insects, like locusts (*von Hadeln et al., 2020*) and butterflies (*Heinze and Reppert, 2012*). As the layers of the CB largely correspond to innervation by tangential input neurons, this suggests that which information is fed into the CX is highly species specific. In contrast, the computational algorithms (defined by the columnar layout) that use this

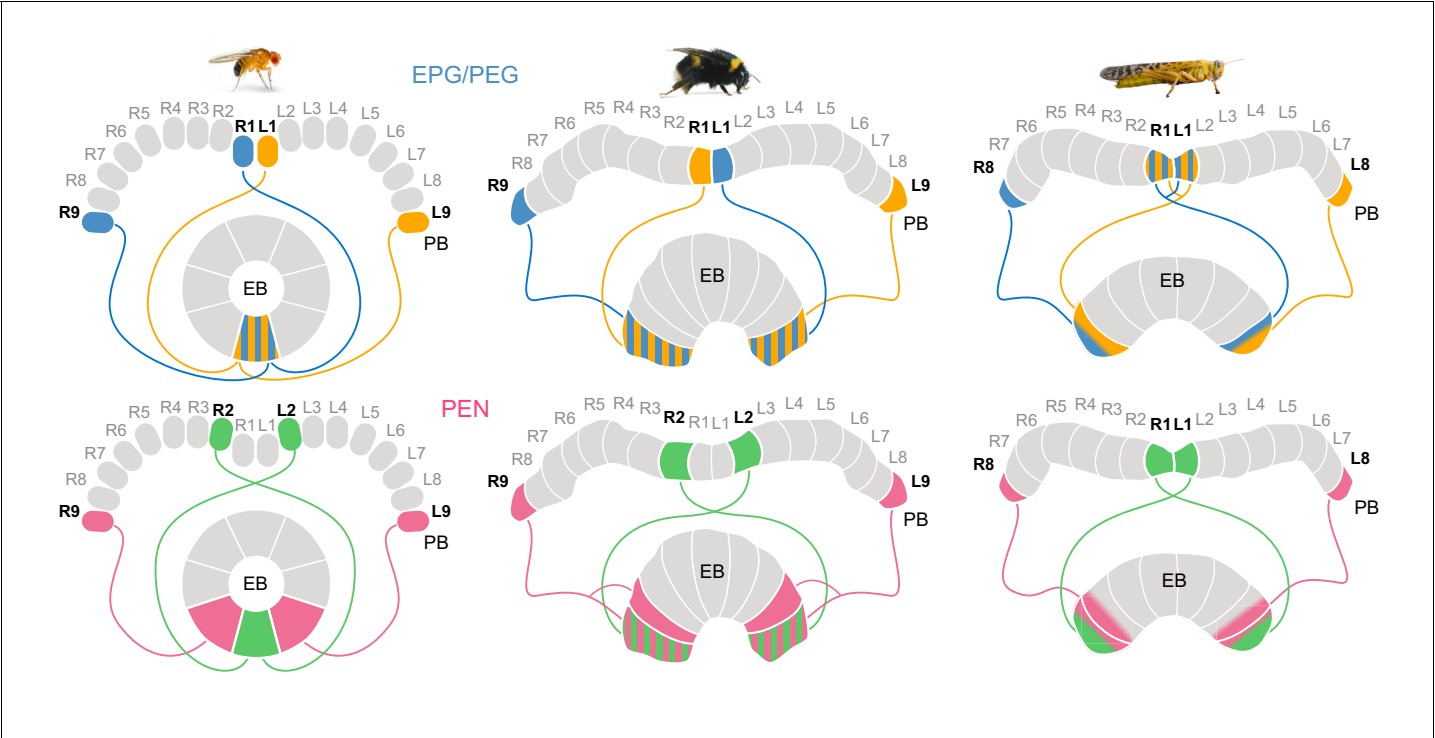

**Figure 12.** Different circuit solutions for anatomically closing the loop in three insect species. Stripped patterns indicate overlapping neural branches. In the fruit fly, EPG cells from medialmost PB columns and EPGt cells from the lateralmost ninth columns have overlapping fibers within the EB, enabling an heading encoding activity bump to move in 360˚. In the bumblebee EB, EPG/PEG cells at the innermost PB column are swapped relative to the fly. Additionally, the projections of the PEN cells in the lateralmost column of the EB cover twice the width as those PEN cells located in all other EB columns. Locusts have evolved a third and slightly different modification, likely a third mechanism for closing the loop (*Pisokas et al., 2020*). They use only eight EPG/PEG neurons instead of nine, but midline crossing input fibers of the innermost EPG cells are suited to functionally close the loop.

input information are much more conserved. These structural differences are consistent with the idea that different sensory cues are relevant to different species , but that the decision about where to turn to in response to sensory information follows the same guiding principles (*Honkanen et al., 2019*). The same argument applies to input into the FB that relays internal state or memory output from the mushroom bodies (*Hulse et al., 2020*).

Finally, the most pronounced difference between bees and flies was found in the organization of the NO. While the fly data, and in fact data on many other species (locusts, butterflies, moths, beetles; *Heinze and Homberg, 2008*; *Heinze and Reppert, 2012*; *Immonen et al., 2017*; *Adden et al., 2020*), reveal a highly structured organization into stacked layers, we found the bee noduli to be comparably disorganized. Our reconstructions only revealed three discrete territories, providing the possibilities for increased levels of cross-talk between the PFN neurons that remain segregated in other species (*Figure 4e-e'*). Interestingly, and discussed in detail below, pronounced intra-columnar microcircuits among PFN cells was one prediction made by *Stone et al., 2017* for insects with highly developed path integration ability, such as bees.

## Conserved core circuits and bee-specific neurons?

In total, we found 11 types of columnar cells (*Figure 3*). Importantly, the majority of these cell types have known homologues in other insects. These include EPG/PEGs, PENs, PFNs, hΔs, and putative motor output cells, PFL1,3 and PFL2 (*Homberg, 1985*; *Heinze and Homberg, 2008*; *Heinze et al., 2013*; *Wolff et al., 2015*; *Stone et al., 2017*; *El Jundi et al., 2018*; *Hensgen et al., 2021*). Drawing comparisons to the fly data, we can distinguish four different categories of cross species resemblance: First, some cell types appear to be extremely conserved, including in quantity, distribution, morphology, and projectivity. In particular, EPG/PEG and PEN cells were indistinguishable between the species with respect to these parameters. Second, other cell types differed in numbers but

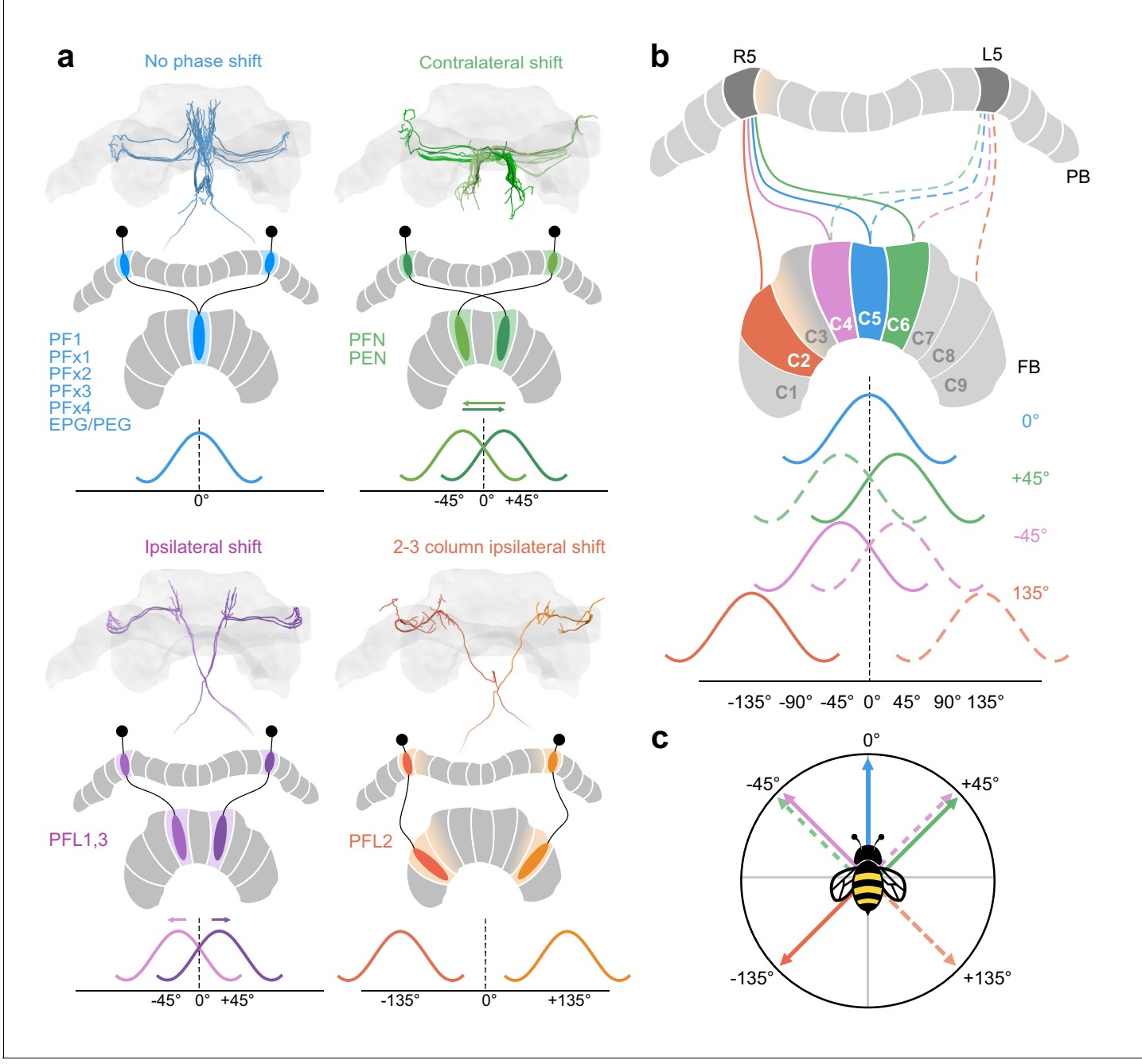

**Figure 13.** Anatomical phase shifts of CX columnar cells. Wave forms assume sinusoidal activity profile as in *Drosophila* (*Hulse et al., 2020*). (a) Most cells follow a default projection pattern whereby there is no offset between their aborizations in the FB and the PB from which they project. PFNs and PENs are offset by one FB column contralaterally resulting in a 45˚ shift relative the default projection pattern. PFL1,3 cells are shifted one column ipsilaterally, also resulting in a 45˚ offset relative to PENs/PFNs, but in the opposite direction. Most PFL2 cells are shifted by three columns ipsilaterally (with the exception of two PFL2 cells, one innervating PB column R3, the other L3; see *Figure 8b*) creating a 135˚ offset relative to the default projection column. Note however, due to overlapping PFL2 branches in the PB a single PFL2 cell may receive input from multiple PB columns (for potential functional implication see *Figure 14—figure supplement 1*). PFL2 cells used as example here have side-branches in PB R5/L5, but are centered in R4/L4 (pastel orange gradient). (b) Each projection offset shown for the cells in (a) projecting from PB columns R5 (cells projecting from L5 indicated by dotted lines but not colored in FB). (c) A vector diagram based on the phase offsets for each cell type, suggesting that each of these cells would be ideal to compare heading signals in the PB with angular signals in the FB at each of the indicated directions.

maintain key characteristics, such as projection pattern, suggesting conserved roles in CX computations. This was the case for PFN and hΔ cells, at least when disregarding potential subtypes. Third, some cell types share quantities, but differ in projectivity. In this category we found the main output neurons of the CX in particular, that is, PFL1,2,3 cells (discussed in detail below). Finally, some cell types appear to have no counterpart in the fly and might be unique to the bee. These include PFx1, PFx2, PFx3, PFx4 (*Figure 10*), PF1 (*Figure 9*), and PFLx neurons (*Figure 3b*), all of which project to the FB following the 'default' pattern.

In contrast to findings in the fruit fly, we did not observe any clear homologues for vΔ cells. In the fly, these cells have input and output branches constrained within single FB columns, but different layers, potentially shifting neural activity vertically (i.e. across layers; *Hulse et al., 2020*). As these neurons have been implied in vector transformations based on fly connectomics data, the lack of such neurons in bees, if confirmed, can be expected to have important functional consequences. However, PF1 neurons in the bee, which only innervate the PB and FB and have branches across multiple layers in the FB may fulfill a similar role. Additionally, bee counterparts of vΔ cells might have small fiber diameters and could have been missed in our analysis.

Interestingly, we also did not find any obvious candidates matching the fly FX cells, columnar neurons lacking arborizations in the PB (*Hulse et al., 2020*). One potential candidate in the bee could be the PFx cells (*Figure 10*). Three of these cell types that have bilaterally projecting fibers toward regions in the superior medial protocerebrum (SMP), similar to the fly FX cells (*Figure 10*). However, whereas FX cells in the fly only arborize in the FB, PFx neurons in the bee also possess arborizations in the PB (*Figure 10*). Intracelluar injection of one such cell revealed processes innervating both hemispheres of regions possibly within the CRE (*Figure 10*; *Figure 10—figure supplement 1c–d'*).

Finally, a fourth FB columnar cell of the bee, PFx2 neurons, sends fibers contralaterally toward the LX, but follows the default projection pattern, making these cells clearly distinct from the PFL output neurons. Given these characteristics, these cells might correspond to *Drosophila* PFRa or PFG cells. Both cells receive input from PFN neurons in flies. This outlines a possible line of enquiry to establish equivalence despite diverging morphology by delineating the up- and downstream synaptic connections and hence place neurons in a corresponding computational context.

## Head direction coding across species

Two columnar cell types, EPG and PEN cells, link the EB and the PB and form the core of the head direction circuit in the fly (*Turner-Evans et al., 2020*). The numbers of both EPG and PEN cells per PB column identified in the bumblebee closely approximated corresponding numbers in the fly (*Figure 3a*, *Figure 3—figure supplement 1*) and, importantly, their projection patterns, characteristically offset by one column in the EB between both cell types, were also conserved (*Figure 6e–f*, *Figure 6—figure supplement 1*).

One major distinction between the bee and the fruit fly is the gross morphology of the EB. As the name suggests, the EB in the fly is toroidal, whereas in the bee the EB more closely resembles an eyebrow or a bent cylinder. In the fly, this morphology is a prime example of structure matching function. Because the EB is shaped as a torus, EPG cells from medialmost PB columns and EPGt cells from the lateralmost ninth columns have overlapping fibers within the EB, enabling an heading encoding activity bump to move in 360° with the same rotational freedom as a compass needle rotating within a compass (*Wolff et al., 2015*; *Hulse et al., 2020*). Based only on gross morphology, the bee EB would not seem to be well suited to serve this same function. However, our data revealed that in bumblebees the proposed head-direction circuit differs from its fly counterpart at lateral extremes of the EB, that is those positions that would correspond to the ring closure in *Drosophila*. Specifically, we found that EPG/PEG cells in the innermost PB columns were swapped in the bee relative to the fly. Unlike in all other columns, these cells project exclusively within the ipsilateral hemisphere and innervate the outermost EB column on the same side of the CX (*Figure 5d,e*; *Figure 6—figure supplement 1*; *Figure 12*). Additionally, the projections of the PEN cells in the lateralmost column of the EB covered twice the width as those PEN cells located in all remaining EB columns. These differences might enable a circuit solution to compensate for the morphologically open EB and produce a closed, ring-like circuit suited to encode smooth rotational movements in the bumblebee.

Intriguingly, in a recent modeling study, *Pisokas et al., 2020* proposed that locusts have evolved a third solution to closing the loop, a solution that differs from both the bee and the fly (*Figure 12*).

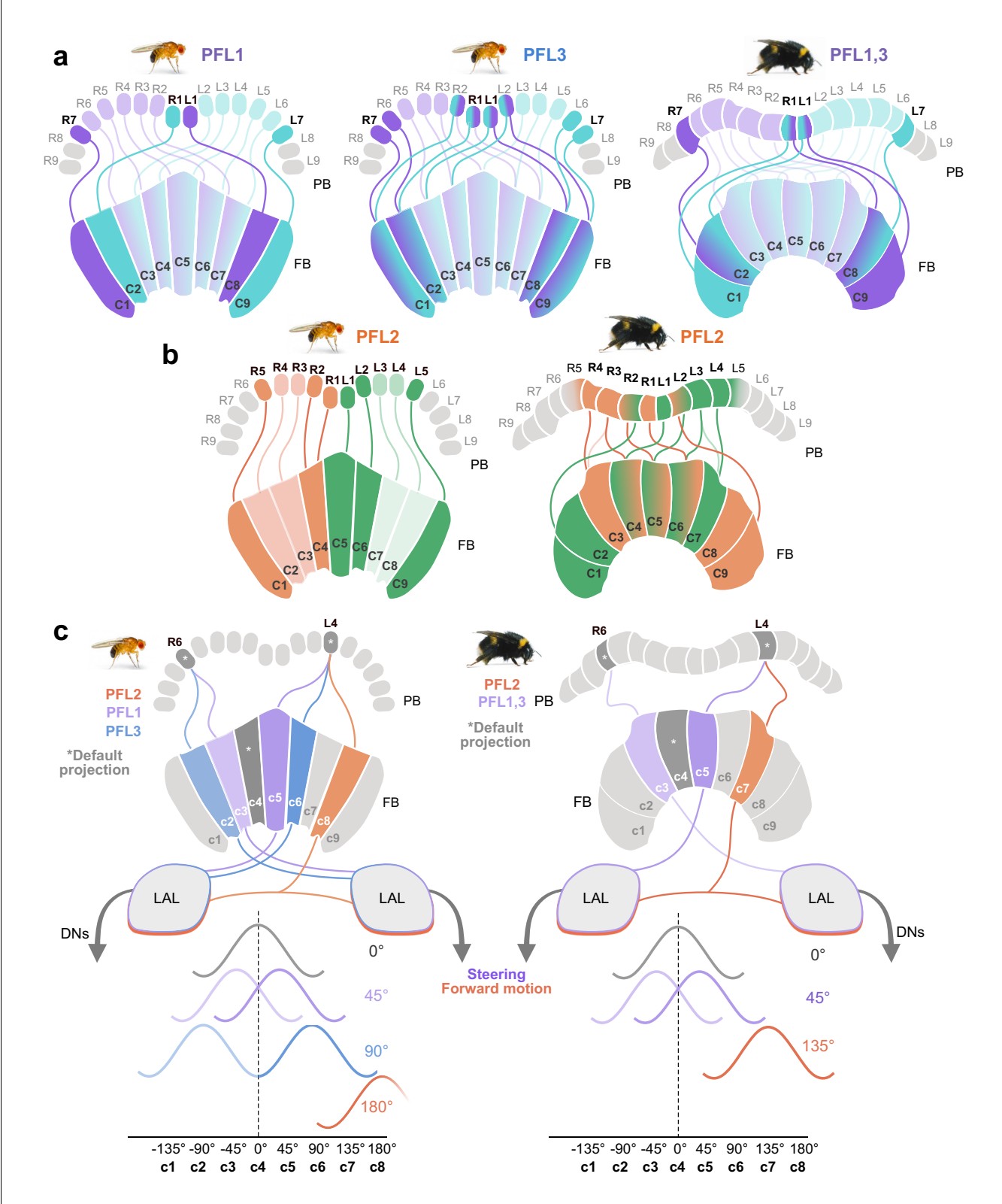

**Figure 14.** Proposed steering CX output neurons (PFL cells) differ in their projection patterns between the bee and the fly. (**a**) Schematic of PFL1 and PFL3 projection patterns in the fly compared to PFL1,3 projection patterns in the bee. In the FB, fruit fly PFL1s are offset by one column ipsilaterally and PFL3s by two. All PFL1,3s in the bee are shifted ipsilaterally by a single column in the FB, with no evidence for a second PFL cell type matching the pattern of PFL3 cells. (**b**) Schematic of PFL2 projections in fly and bee. In contrast to the fly, many bee PFL2 cells have contralaterally projecting PFL2

*Figure 14 continued on next page*

*Figure 14 continued*

cells in the FB which overlap with PFL2s projecting from the opposite hemisphere. Further, PFL2 cells in the fly are ipsilaterally shifted by four columns (180° offset), whereas in bees they are only shifted by three columns (135° offset). (c) Schematic of hypothetical activity profiles that would result from observed differences between fly and bee PFL projection patterns following model proposed by *Hulse et al., 2020*.
The online version of this article includes the following figure supplement(s) for figure 14:

**Figure supplement 1.** PFL2 cells may receive input across multiple PB columns.

While most insects do not have a closed EB, evidence presented here and by *Pisokas et al., 2020* suggests that these different circuit modifications may fulfill the same functional purpose. It is worth noting however, that by modeling the head direction circuit in the locust and the fly, *Pisokas et al., 2020* found that other, more minor, morphological differences between the circuits of both species can significantly alter the dynamics of the ring attractor circuit underlying the head direction system in ways that could adapt circuit function in line with species specific behavioral demands. It will thus be interesting to explore the possible functional consequences and possible behavioral correlates of the detailed characteristics of the bumblebee head direction system.

## Functional implications of the neuroarchitecture of the noduli

Several recent functional studies have implicated the NO as centers that relay idiothetic self-motion cues, including translational optic flow (*Lu et al., 2020*; *Lyu et al., 2020*; *Stone et al., 2017*), rotational angular velocity (*Green et al., 2017*; *Turner-Evans et al., 2017*), and wind direction (*Currier et al., 2020*) to other regions within the CX. So far three neuron types that innervate the NO have been shown to facilitate the encoding of self-motion cues, with anatomical homologues having been identified across insect species (*Heinze and Homberg, 2008*; *Heinze et al., 2013*; *Wolff et al., 2015*; *Stone et al., 2017*; *El Jundi et al., 2018*; *von Hadeln et al., 2020*; *Hensgen et al., 2021*). These include PFN, PEN, and LNO tangential cells (*Figure 4e-f*). LNO tangential neurons provide predominantly input to PFN and PEN neurons, carrying sensory information from other regions of the brain, mostly from the ipsilateral LX (*Stone et al., 2017*; *Currier et al., 2020*; *Lyu et al., 2020*; *Lu et al., 2020*; *Hulse et al., 2020*).

Different LNO types carry sensory information to the NO along multiple parallel channels. In the fly, this is reflected by the structural layout of the NO, in which the fibers of LNO, PFN, and PEN cells are neatly organized into discrete layers. Five subtypes of PFN cells project into five territories in the fly NO. With the addition of PENs that arborize in the dorsalmost layer, there are six distinct zones, each supplied by its own LNO input channel (*Wolff and Rubin, 2018*; *Hulse et al., 2020*). Additionally, the fly NO contains numerous FB tangential cells (FBt) which overlap with and in some cases receive direct input from LNO cells (*Hulse et al., 2020*).

In the bee, we found three anatomical domains, NOs, NOm, and NOc, based on the input arborizations from tangential cells (*Figure 4e-g*). Similar to the fly, PEN cells, likely encoding angular velocity, send their processes to a single zone, the NOs, where they receive isolated tangential cell input and do not overlap with PFN cells (*Figure 4e*). In contrast to flies however, PFNs only arborize in two structurally discrete territories, the NOm and NOc. All PFNs that are associated with the same column in the PB display a surprising amount of overlap in the NO, with many spreading throughout the entire NOm or NOc domain (*Figure 4—figure supplement 1b*). Similarly to the PFN organization, the LNO input cells associated with this region are equally overlapping. Besides suggesting fewer sensory input channels in bees than flies, with the notable exclusion of contralateral input, such an organization additionally offers the possibility of a large degree of intra-column recurrent connectivity in the bee NO relative to the fly, a prediction made by *Stone et al., 2017* to support path integration memory. However, connectivity data will be needed to verify such a claim. A second prediction of this model was that more PFN neurons would be required to increase the capacity and precision of vector memory in species with highly developed path integration. Given that we found more than twice the number of PFN cells in bees compared to the fly, this prediction is clearly met, even though its functional significance remains hypothetical.

Even if all circuits present in flies are conserved in bees, the newly discovered circuit of PFNc neurons in the bee NOc provides an additional possibility for structured recurrent connections between PFN neurons as potential substrate for path integration memory that can be directly used for

steering (*Stone et al., 2017*). The fly connectome data revealed that in *Drosophila* neither highly structured intracolumnar recurrent connections exist, nor are PFN outputs to steering cells hemisphere specific (as required by the *Stone et al., 2017* model). Both assumptions can in theory still be met in PFNc neurons, as long as they receive speed input indirectly, for example via PFNv neurons in the PB.

Another specialization of the bee noduli compared to their fly counterparts is evident in the structure of the NO input. Interestingly, whereas the NOm contains putative input fibers from three LNO cells, the NOc receives no branches from LNO cells at all. Rather, the NOc is innervated by a group of 15–20 FB tangential cells that, based on projection, quantity, and lack of overlap with LNOs, have no equivalent in the *Drosophila* CX and were termed FB-NOc neurons. Based on one completely filled example from *Megalopta*, these cells have distant ventral cell bodies and fibers that extend dorsal-anteriorly, occupying regions in the vicinity of the SMP/CRE (FB-NOc cells; *Figure 4h*, *Figure 4—figure supplement 1d*). While we cannot confirm identical morphologies of the neurons in the bumblebee and these cells might also comprise a family of related cell types rather than a group of identical cells, the fiber trajectories of all found neurons are fully consistent with the morphology of the *Megalopta* neuron. Our data therefore strongly suggest that the FB-NOc tangential neurons provide an additional input channel to the bee NO that specifically synapses onto the PFNc cells, the smallest and most numerous PFN cell type. The much higher cell count of PFN cells in the bumblebee compared to the fly therefore appears to be closely associated with the presence of an additional input channel.

While the function of these FB-NOc tangential cells is unknown, their arborizations in the NO, the FB, and the SMP/CRE and their apparent uniqueness within the bee make them potential candidates to play a role in mediating the highly developed ability of bees for vector navigation. As suggested by *Le Moël et al., 2019*, and based on the computational model by *Stone et al., 2017*, complex navigation strategies, such as trap lining, could be explained by vector navigation, when enabling storage and retrieval of PFN population activity in long-term memory, for example in the mushroom body. Stored vectors from previous foraging segments could be compared to and combined with vectors representing the current state of the path integrator, enabling the choice of optimal foraging routes and novel shortcuts (*Le Moël et al., 2019*). The key prediction of such a model is direct information flow between the mushroom body and the CX path integration memory, for example PFN neurons, both for storing and for recalling vectors. The found FB-NOc neurons are anatomically suited to serve this function, as their branches outside the CX coincide with brain regions innervated by MB output neurons (*Rybak and Menzel, 1993*). A prediction of this hypothesis is that the connection of these neurons to and from PFNc neurons is column specific, an idea that is directly testable via connectomics work and which is consistent with the high number of 15–20 FB-NOc neurons per hemisphere (ca. two per PFN column).

In summary, the specializations identified in the noduli of the bumblebee are consistent with the idea that this region plays a key role in path integration and that the highly developed abilities for path integration related behaviors in bees are indeed reflected in the circuitry of the noduli. Further analysis of the synaptic connectivity of the NOc circuit will have to test these predictions.

## Proposed steering circuits differ between insect species

PFL neurons relay signals to descending neurons in the lateral accessory lobes (LAL), making them likely candidates for cells which propagate steering or other motor commands to the ventral nerve cord (VNC) (*Figure 7*; *Figure 8*; *Stone et al., 2017*; *Steinbeck et al., 2020*; *Rayshubskiy et al., 2020*; *Hulse et al., 2020*). Connectomic analysis of the fly CX revealed the presence of three types of PFL neurons that differ in their PB-FB projection offset and downstream partners. PFL1 cells were found to be offset in their PB-FB projections by one column ipsilaterally, PFL3 cells by two columns ipsilaterally, and PFL2 cells by four columns ipsilaterally (*Hulse et al., 2020*). While PFL1 and PFL3 cells send output fibers to the contralateral LAL, PFL2 neurons contain bilaterally projecting output fibers that arborize in both LALs.

In the bumblebee, we found evidence for a single cell type that shares characteristics from both fly PFL1 and PFL3 neurons (*Figure 7*). Interestingly, bee PFL1,3 neurons only have a one column ipsilateral PB-FB projection offset ('PB-Shifted'; *Figure 7*,*Figure 13*). Intracellular injection of a PFL1,3 cell revealed blebbed varicosities in the LAL and mixed branches in the FB, suggesting that this polarity is maintained across insects (*Figure 7a*; *Hensgen et al., 2021*). Although detailed analysis of

the FB projection fields of these neurons will have to confirm the identical offset patterns for PFL1 and PFL3 cells, our data suggests that the PFL3 projection pattern of the fly does not exist in bees.

Additional discrepancies between bees and flies were identified in PFL2 neurons (*Figure 14*). With a four column ipsilateral PB-FB offset, fly PFL2 neurons project only to ipsilateral regions within the FB. In contrast, bee PFL2 neurons show only a three column ipsilateral offset between PF-FB columnar arborizations ('Both-shifted'; *Figure 8d–e*). Notably, there is one PFL2 neuron in both hemispheres that differs in its projection offset compared to the other PFL2 cells by being shifted one column further ipsilaterally, matching the fly projection offset (*Figure 8b*; R3 and L3). Further, the PFL2 neurons in the lateral most regions appear to project to the two furthest contralateral columns in the FB (*Figure 8d–e*; *Figure 14*), generating an offset pattern that is largely dependent on which PB column an individual PFL2 cell originates in. The very wide input domains in the PB further complicate functional interpretations for PFL2 neurons in bees, as they appear to be active across a much wider range of directions than other cells (each PFL2 cell covers a range of at least 90° of head directions; *Figure 14—figure supplement 1*).

PFL neurons are thought to relay motor commands, such as steering (PFL3 neurons; *Rayshubskiy et al., 2020*; *Hulse et al., 2020*) and forward velocity (PFL2 neurons; *Hulse et al., 2020*), from the CX to the LALs (*Stone et al., 2017*; *Steinbeck et al., 2020*). Given that PFL1,3 neurons in the bee share characteristics with both fly PFL1 and PFL3 neurons, such an arrangement may suggest that the projection offset of these neurons evolutionarily diverged in the fly. This is supported by the fact that similar numbers of cells per column exist for all PFL1 + PFL3 neurons in the fly and all PFL1,3 neurons in the bee. Further, an identical bee-like projection offset for both PFL1 and PFL3 neurons is also consistent with data from dye filled neurons in the Monarch butterfly (*Heinze et al., 2013*). PFL3s in the fly would be well suited to drive rotational movements when the fly is facing +/- 90° away from a goal location (*Hulse et al., 2020*). Due to their offsets in the FB and bilaterally projecting output fibers in the LALs, PFL2s were suggested by *Hulse et al., 2020* to drive forward motion when the fly is facing in the direction opposite to their 'stored' vector (i.e. facing toward their starting point or goal vector).

Assuming a corresponding function in the bee, PFL1,3 neurons would be most active (i.e. propagate steering commands) when bees are facing +/- 45° away from their 'stored' vector and PFL2 neurons would drive forward velocity when they are facing +/- 135° away from their stored vector (although see *Figure 14—figure supplement 1*). Interestingly, if we assume direct and exclusively contralateral connections between PFN and PFL neurons to underlie steering as hypothesized by *Stone et al., 2017*, the opposite offset by PFN neurons would add 45° to these values, leading to a fly-like +/- 90° and +/- 180° away from their 'stored' vector (*Figure 13*). But as the detailed projections of PFL cells, that is the extent of their inputs in the PB and their projection width and innervated layer in the FB, are unknown and partially differ between cells originating in different PB columns, conclusions about phase offsets and possible functional implications for behavior remain preliminary, especially without support from computational models. Nevertheless, the distinct anatomical offsets between flies and bees clearly suggest that bees have evolved a different mechanism for target driven steering, as strictly applying the fly-derived steering concept to bees would result in a 45° error relative to a stored vector. Synaptic resolution connectivity data are clearly required to resolve this interspecies discrepancy.

Overall, the differences found at the level of the main output neurons of the CX are intriguing and suggest that the behavioral control of steering does depend on the species, possibly explaining divergent flight patterns and, combined with distinct sensory input channels, might possibly even account for different navigational strategies.

## Conclusion

In summary, while limited in resolution and missing the smaller cell types of the CX as well as the projections that leave the imaged volume, the projectome analysis of the bumblebee CX provides pioneering insights about the quantity and projectivity of all major CX neurons in the brain of the bumblebee. To the best of our knowledge, this project is the first attempt at comprehensively mapping all columnar and hΔ cells in a non-dipteran insect species, the projections of which define and constrain computations carried out by the CX.

# Materials and methods

## Key resources table

| Reagent type (species) or resource | Designation | Source or reference | Identifiers | Additional information |
|---|---|---|---|---|
| Organism (*Bombus terrestris*) | NA | Koppert, Netherlands | NATUPOL | NA |
| Organism (*Megalopta genalis*) | NA | Wild caught, Barro Colorado Island, Panama | NA | NA |
| Antibody | Synapsin (SYNORF1; Mouse, monoclonal) | Developmental Studies Hybridoma Bank (DHSB) | 3C11 | NA |
| Antibody | Serotonin (5HT; Rabbit, polyclonal) | ImmunoStar | CAT#: 20080; RRID:AB_572263 | NA |
| Antibody | Tyrosine hydroxylase (TH; Mouse, monoclonal) | ImmunoStar | CAT#: 22941; RRID:AB_572268 | NA |
| Antibody | AZniPure Donkey Anti-Mouse IgG (H+L) Cy3 (polyclonal) | Jackson ImmunoResearch | CAT#: 715-165-150; RRID:AB_2340813 | NA |
| Antibody | AZniPure Donkey Anti-Rabbit IgG (H+L) Cy5 (polyclonal) | Jackson ImmunoResearch | CAT#: 711-175-152; RRID:AB_2340607 | NA |
| Other (serum) | Normal goat serum | Jackson ImmunoResearch | CAT#: 005-000-001 | NA |
| Chemical compound | Neurobiotin | Vector Laboratories | CAT#: SP-1120 | NA |
| Chemical compound | Cy3 Streptavidin | Jackson ImmunoResearch | CAT#: 016-160-084 | NA |
| Chemical compound | Permount | Fisher Scienti1 | CAT#: SP15-100 | NA |
| Other (enzyme) | Pronase | Sigma-Aldrich | CAT#: P8811 | NA |
| Chemical compound | Osmium tetroxide 4% aqueous | Proscitech | CAT#: C011 | NA |
| Chemical compound | Potassium ferricyanide | Sigma-Aldrich | CAT#: 702587 | NA |
| Chemical compound | Thiocarbohydrazide | Sigma-Aldrich | CAT#: 223220 | NA |
| Chemical compound | Lead nitrate | Sigma-Aldrich | CAT#: 228621 | NA |
| Chemical compound | L-Aspartic acid | Sigma-Aldrich | CAT#: A4534 | NA |
| Chemical compound | Uranyl acetate | Electron Microscopy Sciences | CAT#: 22400 | NA |
| Chemical compound | Durcupan ACM resin (4-part component kit) | Sigma-Aldrich | CAT#: 44610 | NA |
| Chemical compound | Conductive silver epoxy (2-part) | Ted Pella | CAT#: 16043 | NA |
| Software, algorithm | Amira 5.3 | Thermo Fisher Scienti1c | NA | NA |
| Software, algorithm | Skeletonize plugin for Amira 5.3 | Dr. J.F. Evers *Schmitt et al., 2004* | NA | NA |
| Software, algorithm | CATMAID | *Saalfeld et al., 2009* | NA | NA |
| Software, algorithm | FIJI (ImageJ) | *Schindelin et al., 2012* | NA | NA |
| Software, algorithm | natverse | *Bates et al., 2020* | NA | NA |
| Software, algorithm | Leica Application Suite X (LAS X) Navigator | Leica | NA | NA |
| Software, algorithm | ImageJ image stitching plugin | *Preibisch et al., 2009* | NA | NA |
| Other | Sigma VP 300 SEM | Zeiss | NA | NA |
| Other | 3view 2 system | Gatan | NA | NA |
| Other | Leica SP8 DLS inverted confocal microscope | Leica | NA | NA |

*Continued on next page*

*Continued*

| Reagent type (species) or resource | Designation | Source or reference | Identifiers | Additional information |
|---|---|---|---|---|
| Other | Ampli1er (Panama): BA-03X | NPI | BA-03X | NA |
| Other | Ampli1er (Lund): SEC05-LX | NPI | SEC05-LX | NA |
| Other | Micromanipulator: SMXS-K-R | SensApex OY | SMXS-K-R | NA |
| Other | Electrode puller: P-97 | Sutter | NA | NA |
| Other | Borosilicate glass capillaries w/ filament | Sutter | BF150-75-10 | NA |

## Nomenclature

In past studies, nomenclature used to identify cells and CX areas differed between fruit flies and other insects. Due to overwhelming support for CX homology across insects, the greater availability of functional and neuroanatomical data in *Drosophila* (*Hulse et al., 2020*), and in an effort to make our anatomical descriptions more comparable to the *Drosophila* connectomics data, we adhere here to the short hand nomenclature used for *Drosophila* (*Ito et al., 2014*; *Scheffer et al., 2020*; *Hulse et al., 2020*) but have also provided alternative names historically used in other insects (*Table 1*).

Briefly, columnar neurons are named with a three letter abbreviation according to the neuropils they innervate. In the fruit fly, these abbreviations are ordered by connectivity, with the first two letters being determined by the two neuropils the neuron receives the most input from, in order of which has the greatest amount of input first, and the third letter is given by the neuropil in which the neuron sends predominantly output to. For instance, a PEG cell receives mostly input in the PB and the EB and sends output to the gall. For a more in-depth description of fruit fly CX neuron nomenclature, see *Hulse et al., 2020*. Since connectivity information does not exist for neurons in our dataset, we define neuron homology based on corresponding morphology and arborization domains between species. As general connectivity patterns are likely conserved as well, based on the tight anatomical resemblance as well as functional conservation of CX neurons across species, we have adopted the fly naming scheme for the bumblebee. For neurons in which there are no clear homologues in the fly, we used the neuropils that they innervate within the imaged volume to define the neuron name. For example, PFx neurons innervate the PB, the FB, and an unknown tertiary region denoted here as 'x'. Lastly, we consider all CX neurons to fall into one of three classes: columnar neurons, FB interneurons (i.e., hΔ), and tangential neurons (see Introduction). Neurons within these classes were defined as belonging to the same neuron type if they were anatomically indistinguishable from each other, yielding the columnar neuron types listed in *Table 1* as well as several types of hΔ neurons. While some of these likely consist of several distinct subtypes based on connectivity and detailed projection areas (e.g. PFN neurons), we were not able to unambiguously identify those due to the limits imposed by the resolution of our image data.

## Animals

Bumblebees (*Bombus terrestris*), were obtained from a commercial supplier (Koppert, Netherlands) and kept in a greenhouse on campus at Lund University, Sweden. Bee colonies were maintained inside room-sized flight canvases with sugar water and pollen available from artificial feeders at all times. Only adult female workers were used in this study.

A few of the intracellular dye injections were performed on female adult sweat bees (*Megalopta genalis*) and were used in this study to confirm single-cell morphologies where possible. Sweat bees were collected in the rainforest of Barro Colorado Island (field station of the Smithsonian Tropical Research Institute, Panama). Sweat bees were collected with a light trap made using a white sheet and illuminated by a light source containing UV wavelengths. Traps were set during early morning twilight, a time when the bees are most active. Captured bees were kept in vials containing containing honey solution and water solution soaked into cotton balls and were processed within 2 weeks following capture.

## Immunohistochemistry

Whole mount synapsin immunolabeling followed a slightly modified version of the method described by *Ott, 2008*. Bees were anesthetized to immobility over ice, after which their heads were removed and immediately placed into freshly made fixative containing 1% paraformaldehyde (PFA), 0.25% zinc chloride, 0.79% sodium chloride, and 1.2% sucrose. Neural tissue was dissected free from the head capsule, removing as much of the neural sheath as possible, and left to fix overnight at 4°C.

The next day, brain tissue was rinsed 8 x with HEPES buffered saline (HBS) over the course of an hour. Tissue was then rinsed 3 x for 10 min intervals in Tris-HCl (pH 7.3) before being transferred into Dent's fixative containing 80:20 methanol to dimethyl sulfoxide (DMSO) were it was left to soak for 75 min on a gentle shake. Next, neural tissue was rinsed 3 x in Tris-HCl over 10 min intervals and placed into a blocking solution consisting of 5% normal donkey serum (NDS) in 0.01 M phosphate buffered saline with 0.5% Triton-X 100 (PBST). After a 3 hr incubation, blocking solution was swapped with primary antibody solution containing 1% NDS, 0.5% PBST, and 1:25 mouse anti-synapsin monoclonal antibodies (3C11; Developmental Studies Hybridoma Bank; Iowa City, IA). In some experiments, a polyclonal antibody raised in rabbit against serotonin (5-HT; AB_572263; ImmunoStar; Hudson, WI) was also included at a concentration of 1:2000. Brains were left to incubate in this solution either in the fridge or at room temperature on a shaker for 4–6 days.

The following day, brains were rinsed 6 x over the course of an hour in 0.1% PBST. Meanwhile, whole IgG secondary antibodies raised in goat and conjugated to Cy3 or Cy5 (AB_2338003 and AB_2338713, respectfully; Jackson ImmunoResearch; Philadelphia, PA) were added to Eppendorf tubes containing 1% PBST at a concentration of 1:300. Anti-mouse conjugated to Cy5 was used to detect synapsin and anti-rabbit conjugated to Cy3 to detect 5-HT, when applicable. Secondary-containing Eppendorf tubes were then placed in a centrifuge and spun for 5–10 min at 8000 g to separate potential antibody aggregates. The top 900 µl of secondary antibody solution was then added to the whole brains, which were subsequently left to incubate for 1–3 days either at 4°C, or on a shaker at room temperature.

After secondary antibody labeling, brain tissue was washed 3 x 20 min with 0.1% PBST, 2 x 10 min with PBS, and then dehydrated through a series of increasing ethanol concentrations (50%, 70%, 90%, 2 x 100%) at 10 min intervals. Brains were next transferred into a 1:1 mixture of ethanol to methyl salicylate for 15 min, and then cleared in 100% methyl salicylate for 75 min. Finally, brains were mounted in Permount mounting medium on slides between two coverslips and separated by spacers, where they were stored until imaging.

For tyrosine hydroxylase (TH) immunolabeling, brains were dissected and fixed in 4% PFA in 0.1 M PBS containing 3% sucrose for 30–45 min. Fixation times lasting longer than an hour resulted in reduced or completely absent immunoreactivity, as has been previously reported in locusts (*Lange and Chan, 2008*). Whole brains were then rinsed 2 x 10 min in 0.1 M PBS, 3 x 10 min in 1% PBST, and subsequently blocked in 1% PBST containing 5% NGS for 3 hr on a gentle shake. Next, neural tissue was left to incubate for 2–3 days in primary antibody solution consisting of 1:250 mouse anti-TH (AB_572268; ImmunoStar; Hudson, WI) in 1% PBST with 1% NGS. Rinsing, secondary antibody labeling, dehydration, and mounting were then performed as described above.

## Intracellular dye injections

Intracellular dye injections were carried out during intracellular electrophysiology and followed the protocol described in *Stone et al., 2017*. In short, electrodes with a resistance of 50–150 MΩ were drawn from borosilicate glass capilaries (Sutter P-97 puller). Bees were anesthetized over ice until immobile and then waved to a plastic holder. A frontal window was cut into the head cuticle and, where necessary, air sacs, fat, and neural sheath were removed using tweezers. Electrode tips were then filled with 4% neurobiotin (Vector Laboratories) in 1 M KCl, and backed with 1 M KCl. Silver wire placed in the ventral part of the head near the mandibles as a reference electrode and the main electrode inserted frontally in the brain and positioned using a micromanipulater (Sensapex, stepping mode). Once a cell was successfully impaled, a depolarizing current of 1–3 nA was applied to iontophoretically inject neurobiotin.

Injected brains were dissected free from the head capsule and fixed in a solution containing 4% PFA, 2% saturated picric acid, and 0.25% glutaraldehyde in 0.01 M PBS overnight at 4°C. The next morning, brains were washed 4 x 15 min in 0.01 M PBS and then left to incubate in 0.3% PBST

containing 1:1000 Streptavidin conjugated to Cy3 for 3 days. Next, the brains were washed 4 x 20 min in 0.3% PBST and dehydrated in an increasing ethanol series, cleared in methyl salicylate and mounted in Permount as described in the previous section.

Mass dye applications followed an identical protocol, but instead of intracellular iontophoretic injection, neurobiotin crystals were applied directly to the tissue. To that aim, the fine tip of an intracellular recording electrode was broken off and the remaining, coarse tip was dipped in petroleum jelly. The coated tip was used to pick up a few crystals of neurobiotin and then manually inserted into the target region of the exposed brain. Before injection, the brain was desheathed and all overlaying liquid was removed. After injection and careful rinsing with bee ringer solution, the dye was allowed to diffuse for 30 min, before the brain was removed from the head capsule and fixated in 4% PFA.

To better determine location of arborization domains, injected brains were imaged, re-hydrated, and co-stained with immunohistochemical markers using the method reported by *Heinze et al., 2013*. Briefly, brain tissue mounted in Permount was freed by soaking in xylene for 2–3 hr before rehydration in a decreasing ethanol series of 100% x 2, 90%, 70%, and 50% ethanol at 15 min steps. Brains were then rinsed 3 x 15 min in 0.01 M PBS and transferred to 0.5% PBST. Next, brains were embedded in albumin gelatin (4.8% gelatin and 12% ovalbumin) and post fixed with 4% formalin overnight at 4°C. The following day, brains were rinsed in 0.01 M PBS 3 x 15 min and vibratome sectioned at 140 μm. They were subsequently processed for immunohistochemistry as described in the previous section beginning at the blocking step, although with shorter incubation periods. Brain sections were left in blocking solution for 3 days, primary antibody solution for 4 days, and secondary antibody solution for 2 days. Rinse cycles and temperatures were the same. Following antibody incubation, sections were rinsed 4 x 15 min in 0.1% PBST, 2 x 15 min in 0.01 M PBS, then dehydrated in an increasing ethanol series at 10 min steps, cleared in methyl salicylate for 20 min and mounted in Permount between two coverslips separated by spacers.

## Serial block-face electron microscopy

Methodology used here has been previously reported in *Stone et al., 2017*. Briefly, bee brains were dissected and fixed in a solution of 4% PFA and 2% glutaraldehyde in sodium cacodylate buffer overnight at 4°C. Neural tissue was then rinsed 4 x 15 min in 0.01 M PBS, embedded in albumin/gelatin, and post fixed overnight at 4°C. To image the CX, a single thick section (400 μm) was cut from the albumin/gelatin block using a vibrating blade microtome and stored in 0.01 M PBS until further processing. This same technique was used for the noduli, using a smaller section thickness (200 μm). Large volume en bloc staining was then performed, beginning with osmification in a solution containing 2% osmium tetroxide and 1.5% potassium ferricyanide in double distilled water (ddH20) for 1 hr at room temperature. Tissue was then washed 3 x 5 min in ddH20 and then sequentially immersed in 1% thiocarbohydrazide for 20 min and 2% osmium tetroxide for 30 min both steps being followed by 3 x 5 min rinses with ddH20. Neural tissue was then left to incubate in 1% uranyl acetate overnight at 4°C. The next day, tissue was washed 3 x 5 min with ddH20 before being left to soak in lead aspartate for 60 min at 60°C. Lead aspartate solution was made by adding 0.066 g lead nitrate to 10 ml ddH20, with a pH adjusted to 5.5 using KOH. Next, tissue was rinsed 3 x 5 min with ddH20 and dehydrated in an increasing ethanol series (20%, 50%, 70%, 90%, 2 x 100%) at 5 min intervals. Samples were then slowly infiltrated with increasing concentrations of Durcupan resin to ethanol (25%, 50%, 75%) at 2 hr intervals and left in 100% Durcupan overnight. The following day, tissue was transferred to fresh Durcupan for 2 hr and left to polymerize for 48 hr at 60°C. Finally, samples were trimmed and mounted onto aluminum stubs using two part conductive silver epoxy.

## Image acquisition and processing

Blocks of brain tissue were imaged using a Zeiss Sigma VP scanning electron microscope equipped with a Gatan 3View ultramicrotome. Three different scans were acquired for this study: an overview scan of the entire central complex imaged at a voxel size of 126 nm x 126 nm x 100 nm (field of view 400 μm x 400 μm), a slightly higher resolution scan of the noduli at 100 nm x 100 nm x 100 nm (field of view 95 μm x 95 μm), and a high-resolution scan of the noduli at 23.6 nm x 23.6 nm x 50 nm (field of view 46 μm x 46 μm). All scans were obtained with a beam energy of 2 kV under high vacuum. Following image acquisition, image alignment and contrast optimization were carried out using

Amira 5.3. Image stacks were then down sampled to 8-bit depth, enabling the use of the Skeletonize plugin for Amira (*Schmitt et al., 2004*) as well as to perform image segmentation using Amira's segmentation editor. For segmentation of CX neuropils in each of the three SBEM data sets the image stacks were down-sampled to in Amira to 1 μm x 1 μm x 1 μm voxel size.

A Leica SP8 DLS inverted confocal microscope was used to collect image data from immunohistochemically labeled brains as well as for individual neurons injected with neurobiotin. Image stacks of entire brains were generated using a 20x oil-immersion objective. To accomplish this, a mosaic of 8–10 image stacks (voxel size of 0.76 μm x 0.76 μm x 1 μm) were collected and stitched together in the x-y plane using Leica Application Suite X (LAS X) Navigator. Due to the relatively large brain size and limited working distance, brains had to be imaged both frontally and posteriorly with some overlap to allow for subsequent merging. Frontal and posterior image stacks were manually positioned in Amira using the transform tool and then registered using the affine registration tool with one stack as a reference. Once the transformations were applied to the stack being registered, both were exported and stitched together to form one stack of the full brain using the stitching plugin in FIJI (*Preibisch et al., 2009*; *Schindelin et al., 2012*). The resulting image stack was then down-sampled in Amira to 1 μm x 1 μm x 1 μm voxel size to enable reconstruction using Amira's segmentation editor.

## Segmentation and neuron tracing

Neurons were manually reconstructed for all three SBEM data sets using both Amira (Thermo Fisher Scientific) and CATMAID software (*Saalfeld et al., 2009*). We used Amira with the third-party plugin Skeletonize (*Schmitt et al., 2004*) at the beginning stages of this project. With this method, branch points (nodes) are placed along the length of a neuronal process. Nodes are automatically connected by a straight line (edge), the diameter of which can be adjusted to approximate the diameter of the neuronal process that is being traced. Amira enabled the use of orthogonal slices along the xz and yz planes which contained a better resolution in the lowest resolution data set (126 nm x 100 nm compared to 126 nm x 126 nm). The Skeletonize plugin was also used to reconstruct and visualize neurons injected with neurobiotin.

At later stages in the development of this study, we transferred all tracing data from Amira to CATMAID, an open-source and collaborative neural reconstruction software that enabled us to simultaneously trace EM image data over the web (*Saalfeld et al., 2009*). We did so with the aid of natverse, a suite of R packages that include many functions for handling and analysing neuroanatomical data (*Bates et al., 2020*). We specifically used the nat and catmaid packages to convert, rescale, and upload neurons and neuropil surface data from Amira to CATMAID. Once in CATMAID, neuron skeletons were manually traced by placing nodes along the length of a neuronal fiber. CATMAID's 3D Viewer was used to visualize neurons and generate many of the images used throughout this study. To increase tracing efficiency, we did not adjust node diameter to approximate fiber diameter when tracing in CATMAID as was done in Amira.

Neuropils were reconstructed using the segmentation editor in Amira 5.3. This was carried out by tracing key cross-sections in all three spatial planes using the paintbrush tool. Such a process results in a scaffold which was then turned into a surface by interpolation using the 'wrap' function. This method was used to reconstruction neuropils in each of the three SBEM data sets as well as whole mount brains immunolabeled with synapsin, serotonin, or tyrosine hydroxylase.

Immunolabeling of serotonin and tyrosine hydroxylase was used to aide in the visualization of layers within the CB (available in the insect brain database, Heinze et al.,2021). Using Amiras segmentation editor, immunoreactive layers in the CX were reconstructed for each immunostained brain and were affine registered to the CB of the EM data. Doing so enabled us to establish a rough idea of layering within the FB and EB, and to visualize the trajectories of reconstructed neurons from the 126 nm data set relative to these immunopositive layers.

## Data availability

Reconstructed neurons are available to view and download on the Insect Brain Database (IBdb; *Heinze et al., 2021*): https://www.insectbraindb.org/app/connectomics;experiment=61;handle=EIN-0000061.1 and https://www.insectbraindb.org/app/connectomics;experiment=62;handle=EIN-0000062.1.

## Acknowledgements

We thank the facilities, and the scientific and technical assistance, of the Microscopy Australia Facility at the Centre for Microscopy and Microanalysis (CMM), The University of Queensland. We are especially grateful to Robyn Chapman, Rick Webb, and Roger Wepf for valuable insights and access to the EM facility. We thank Katinka de Nijs for performing the intracellular dye injections used in this study, Hamza Ali for tracing FB-NOc neurons in the 24 nm data set, as well as Alex Bates and Gregory Jefferis for helpful discussion on analysis and support with natverse. We are grateful to Kevin Tedore (company: 'Kevin Tedore Interactive') for administration of CATMAID and for providing ongoing technical support. Further CATMAID support was provided by Tom Kazimiers. This work was funded by the European Research Council (ERC) under the European Union's Horizon 2020 research and innovation program (to SH, grant agreement no. 714599), the Swedish Research Council (Vetenskapsrådet, to SH, grant agreement no. 2018–04851 and 621-2012-2213), as well as from the Australian Research Council, the Australian Research Foundation and Air Force Office for Scientific Research (AFOSR) to Justin Marshall (to support RT). MES was funded by an International Cotutelle Macquarie University Research Excellence Scholarship (iMQRES 2019060).

## Additional information

### Funding

| Funder | Grant reference number | Author |
| --- | --- | --- |
| H2020 European Research Council | 714599 | Stanley Heinze |
| Swedish Research Council | 2018-04851 and 621-2012-2213 | Stanley Heinze |
| Australian Research Council | | Rachel Templin |
| Australian Research Foundation | | Rachel Templin |
| Air Force Office of Scientific Research | | Rachel Templin |
| Macquarie University | iMQRES 2019060 | Marcel Ethan Sayre |

The funders had no role in study design, data collection and interpretation, or the decision to submit the work for publication.

### Author contributions

Marcel Ethan Sayre, Data curation, Formal analysis, Validation, Investigation, Visualization, Methodology, Writing - original draft, Writing - review and editing; Rachel Templin, Investigation, Methodology, Writing - review and editing; Johanna Chavez, Julian Kempenaers, Investigation, Writing - review and editing; Stanley Heinze, Conceptualization, Resources, Data curation, Supervision, Funding acquisition, Validation, Investigation, Methodology, Project administration, Writing - review and editing

### Author ORCIDs

Marcel Ethan Sayre ![ORCID] https://orcid.org/0000-0002-2667-4228
Rachel Templin ![ORCID] https://orcid.org/0000-0003-2800-5826
Johanna Chavez ![ORCID] https://orcid.org/0000-0003-1366-1409
Julian Kempenaers ![ORCID] http://orcid.org/0000-0002-0059-1045
Stanley Heinze ![ORCID] https://orcid.org/0000-0002-8145-3348

### Decision letter and Author response

Decision letter https://doi.org/10.7554/eLife.68911.sa1
Author response https://doi.org/10.7554/eLife.68911.sa2

## Additional files

### Supplementary files
- Transparent reporting form

### Data availability
Neuron morphologies presented in this paper have been deposited as interactive datasets in the InsectBrainDatabase with accession numbers EIN-0000061 (126nm data) and EIN-0000062 (24nm data). These are available for interactive viewing as well as download.

The following datasets were generated:

| Author(s) | Year | Dataset title | Dataset URL | Database and Identifier |
|---|---|---|---|---|
| Heinze S, Sayre ME, Templin R, Chavez J, Kempenaers J | 2021 | A projectome of the bumblebee central complex - 126nm data | https://hdl.handle.net/20.500.12158/EIN-0000061.1 | Insect Brain Database, EIN-0000061 |
| Heinze S, Sayre ME, Templin R, Chavez J, Kempenaers J | 2021 | A projectome of the bumblebee central complex - 24 nm data | https://hdl.handle.net/20.500.12158/EIN-0000062.1 | Insect Brain Database, EIN-0000062 |

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
