## [Decision Letter]

**Acceptance summary:**

To remark once again, and congratulate your team, on how accessible the data is. Truly a resource to be cherished and shared widely.

**Decision letter after peer review:**

Thank you for submitting your article "A projectome of the bumblebee central complex" for consideration by *eLife*. Your article has been reviewed by 3 peer reviewers, including Albert Cardona as Reviewing Editor and Reviewer #1, and the evaluation has been overseen by K VijayRaghavan as the Senior Editor. The following individual involved in review of your submission has agreed to reveal their identity: Hannah Haberkern (Reviewer #2).

Essential revisions:

1) Please consider implementing some of the suggestions from Reviewer #2 regarding additional references to figures in the main text, with panel-lettering resolution where appropriate, to better follow the text.

2) Please give the discussion a thorough read and consider shorting some bits, perhaps pruning away some of the more speculative statements. This is entirely up to you, in the spirit of making the manuscript more readable to better reach future researchers who may devise experiments to test the formulated hypotheses deriving from this anatomical and comparative study.

*Reviewer #1 (Recommendations for the authors):*

I have not found minor points to enumerate. As far as I can tell, the authors have very carefully checked the numbering and references to figures in the text, as well as language, grammar, and clarity of exposition. I find the figures complete and largely self-explanatory.

*Reviewer #2 (Recommendations for the authors):*

Overall, I found the manuscript easy and enjoyable to read. The figures are beautiful, and the text is well written. I do have a few suggestions that I think would enhance the clarify and presentation further.

Broader points:

1. The text flow occasionally feels disconnected from the figures. I had the impression that the figures as well as the text were each constructed to work well on their own, but on several occasions, I had trouble relating statements in the text to the respective figures. I think two things would help here:

a. Make sure to make frequent references to individual panels rather than going several sentences and the referencing a full figure. This should be the case for the results and the discussion. One example is in the Discussion section, lines 594ff, where it would be nice to refer back to the figures that show the corresponding results. Another example is the paragraph in line 361ff.

b. Revisit the order of the figures and panels. In some cases, I found myself going back and forth a lot, which felt disruptive in my reading process. One example is Figure 5, which does not seem to be well integrated at all. I think one solution could be to move Figure 5C to the SI and then combine Figure 5 and 6.

2. It is not fully clear to me how information from the different datasets was combined. In particular, how did you arrive at the cell number counts for a given type?

3. Further, I would find it helpful to have a "algorithm" of how you arrived at a type label for a given traced neuron. I think it might also be worth discussing the term "type" here a little bit, since the types you consider here are broader and based on less complete information than the types defined in the fly (connectome) and you relate the two. I had trouble understanding how for example PFx and PFL were distinguished.

4. Maybe it would be helpful to indicate where neurites were lost. I found myself wondering this for example in case of the tracings in the FB. It is hard to see this from the figures directly.

5. Some parts of the discussion are a bit too speculative for my taste but maybe I simply had trouble following the logic. It was, for example, not clear to me what made the PFNc types so special (e.g. line 661ff). I felt like some key information, mostly about the connectivity of PFNs in the FB but also in the NO, is missing and I would not focus on discussion topics that sort of require this information to be appropriately addressed. Therefore, my suggestion would be to drastically shorten the Discussion section on "Functional implications of the neuroachitecture of the noduli".

*Reviewer #3 (Recommendations for the authors):*

The description of the CX is quite extensive and informative for the CX aficionado while it may seem a bit hard to read for persons more distant to the topic. However, I am not sure whether I want to recommend shortening – the details may turn out to be important for readers according to their specific interests.

The discussion of the conserved and diverged aspects may profit from shortening (something like 20%?).

The only typo I found:

Line 299…entirely absent PEN neurons….did you mean "devoid of"?

---

## [Author Response]

Essential revisions:1) Please consider implementing some of the suggestions from Reviewer #2 regarding additional references to figures in the main text, with panel-lettering resolution where appropriate, to better follow the text.

We have integrated additional references to figures and specific panels where possible throughout the text. These (as all other changes) have been listed in the tracked changes. We hope that this has integrated the figures better with the text and will provide a smoother experience for the reader.

2) Please give the discussion a thorough read and consider shorting some bits, perhaps pruning away some of the more speculative statements. This is entirely up to you, in the spirit of making the manuscript more readable to better reach future researchers who may devise experiments to test the formulated hypotheses deriving from this anatomical and comparative study.

We have now shortened the discussion at various points to skip some of the intricate detail that might have been distracting for the more general reader. This particularly refers to the section covering the noduli, which was disproportionately long in the original version of the manuscript. As some of the details will be relevant to specialist readers, we have integrated the eliminated sections in a shorter form with the figure legends of the supplemental figures (which had been heavily referenced in the removed sections).

Reviewer #2 (Recommendations for the authors):Overall, I found the manuscript easy and enjoyable to read. The figures are beautiful, and the text is well written. I do have a few suggestions that I think would enhance the clarify and presentation further.Broader points:1. The text flow occasionally feels disconnected from the figures. I had the impression that the figures as well as the text were each constructed to work well on their own, but on several occasions, I had trouble relating statements in the text to the respective figures. I think two things would help here:a. Make sure to make frequent references to individual panels rather than going several sentences and the referencing a full figure. This should be the case for the results and the discussion. One example is in the Discussion section, lines 594ff, where it would be nice to refer back to the figures that show the corresponding results. Another example is the paragraph in line 361ff.

We have updated in-text figure references so that they point to specific panels rather than entire figures as suggested (lines 594 and 361 were included; see revision tracking document for all line numbers).

b. Revisit the order of the figures and panels. In some cases, I found myself going back and forth a lot, which felt disruptive in my reading process. One example is Figure 5, which does not seem to be well integrated at all. I think one solution could be to move Figure 5C to the SI and then combine Figure 5 and 6.

We have checked the order of all panels and figures throughout the manuscript and have now ensured that references to panels reflect the order of these panels in the figures as much as possible. We have left the figures in order as they are, but have worked to better integrate them with the text by supplementing the text with additional references and re-ordering references where possible.

2. It is not fully clear to me how information from the different datasets was combined. In particular, how did you arrive at the cell number counts for a given type?

Most total cell estimates listed in Figure 3 were taken from the 126 nm dataset, with the exception of the two noduli-innervating cell types, PFN and PEN. For these two cells, we derived total numbers from the 24 nm noduli data set, the resolution of which allowed us to trace finer fibers. We assumed bilateral symmetry for these two cell types and therefore included double the number of cells reconstructed from the 24 nm dataset as our estimate. The exact number of cells identified for each hemisphere and from both datasets can be found in Figure 3—figure supplement 1. Figure 3 legend has been updated for clarity.

3. Further, I would find it helpful to have a "algorithm" of how you arrived at a type label for a given traced neuron. I think it might also be worth discussing the term "type" here a little bit, since the types you consider here are broader and based on less complete information than the types defined in the fly (connectome) and you relate the two. I had trouble understanding how for example PFx and PFL were distinguished.

In the Results subsection “Columnar cell types” we state the following about identifying neuronal types:

“We identified columnar cell types based on their projection fields within CX neuropils, their heterolateral projection patterns in the CB (i.e., the column they project to relative to the bundle they originate from), their neurite paths, their population size (number of cells per PB column), and by the tertiary neuropil they innervate.”

In the example of PFx cells and ‘PFL1,3’ cells, ‘PFL1,3’ cells have a single column ipsilateral projection offset in the FB, whereas PFx cells follow the ‘default’ projection pattern. Additionally, PFx neurons follow a different trajectory through the FB compared to PFL neurons, and also have a much thinner main neurites (especially near the PB, where PFL neurons are the largest of all columnar cells).

Because our data sets lack connectivity and full arborization information, we were unable to classify neuronal subtypes (with the exception of PFN main vs PFNc cells, however those likely contain subtypes themselves). However, many neuron types can readily be identified by comparing their morphology and projectivity to light microscopy data of homologous neurons in other species, as was done here with the sweat bee.

To more clearly recapitulate our approach to naming neurons and defining cell types, we have rewritten and expanded the “Nomenclature” section in Methods lines 792ff:

“… Briefly, columnar neurons are named with a three letter abbreviation according to the neuropils they innervate. In the fruit fly, these abbreviations are ordered by connectivity, with the first two letters being determined by the two neuropils the neuron receives the most input from, in order of which has the greatest amount of input first, and the third letter is given by the neuropil in which the neuron sends predominantly output to. For instance, a PEG cell receives mostly input in the PB and the EB and sends output to the gall. For a more in-depth description of fruit fly CX neuron nomenclature, see Hulse, 2020.

Since connectivity information does not exist for neurons in our dataset, we define neuron homology based on corresponding morphology and arborization domains between species. As general connectivity patterns are likely conserved as well, based on the tight anatomical resemblance as well as functional conservation of CX neurons across species, we have adopted the fly naming scheme for the bumble bee. For neurons in which there are no clear homologues in the fly, we used the neuropils that they innervate within the imaged volume to define the neuron name. For example, PFx neurons innervate the PB, the FB, and an unknown tertiary region denoted here as "x". Lastly, we consider all CX neurons to fall into one of three classes: columnar neurons, FB interneurons (i.e., hΔ), and tangential neurons (see Introduction). Neurons within these classes were defined as belonging to the same neuron type if they were anatomically indistinguishable from each other, yielding the columnar neuron types listed in table1 as well as several types of hΔ neurons. While some of these likely consist of several distinct types based on connectivity and detailed projection areas (e.g. PFN neurons), we were not able to unambiguously identify those due to the limits imposed by the resolution of our image data.“

4. Maybe it would be helpful to indicate where neurites were lost. I found myself wondering this for example in case of the tracings in the FB. It is hard to see this from the figures directly.

The endpoints of the neurites are where the neurites were lost and could not be traced further. While it would be possible to highlight these endpoints in the figures, we feel that this would create clutter rather than being helpful. We have thus decided to leave the images as they are for clarity. Generally, fibers that were thinner than 600 nm in diameter were untraceable. Sometimes neurons were also lost due to other issues with the image data. For instance, there may be an image in which a piece of debris from the cutting knife obscured the imaging plane (a common issue in serial-block face EM).

We would like to emphasize at this point, that the datasets underlying this paper are publicly available to view (and download) at the Insect Brain Database (IBdb) [https://www.insectbraindb.org/app/connectomics;experiment=61;handle=EIN-0000061.1 and https://www.insectbraindb.org/app/connectomics;experiment=62;handle=EIN-0000062.1 ]. We have generated a user-friendly 3D viewing platform that will allow users to view the entire data set, search for neuron types, or filter the data according to figures of this manuscript. The interactive viewer is continuously optimized and expanded to obtain a flexible, fast, and aesthetically pleasing rendering of the data (currently optimized for display speed).

Importantly, in this tool, all neurons can be individually assessed by the user. This will allow each reader of the paper to evaluate the data and clearly see the extent to which each neuron could be traced.

5. Some parts of the discussion are a bit too speculative for my taste but maybe I simply had trouble following the logic. It was, for example, not clear to me what made the PFNc types so special (e.g. line 661ff). I felt like some key information, mostly about the connectivity of PFNs in the FB but also in the NO, is missing and I would not focus on discussion topics that sort of require this information to be appropriately addressed. Therefore, my suggestion would be to drastically shorten the Discussion section on "Functional implications of the neuroachitecture of the noduli".

As discussed in lines 647ff, there are several features that make PFNc / FB-NOc cells particularly interesting. To begin with, at 15-20 cells per hemisphere, FB-NOc cells are more numerous than all FBt-NO cell types in a single hemisphere of the fly combined (13; Hulse et al., 2020). Further, FB-NOc cells are not the only tangential cells innervating the NO, as we found six other FBt-NO cells exist alongside FB-NOc cells in the bee. Secondly, they occupy a layer which receives no input from GLNO/LNO cells. The only cells inverating this region (NOc) are PFNc cells and FB-NOc cells. Additionally, based on morphology and arborization domains, there is no obvious corresponding neuron to FB-NOc cells in the fly. All of these features make this system of cells unique to the bumblebee and therefore a target of attention for speculation and future work.

We have now shortened the discussion on the NO and moved some of the more speculative ideas to the Supplemental Figure. We are forthcoming about the speculative nature of these ideas but feel that they are important for guiding future research, so we’ve decided against removing them entirely.

Reviewer #3 (Recommendations for the authors):The description of the CX is quite extensive and informative for the CX aficionado while it may seem a bit hard to read for persons more distant to the topic. However, I am not sure whether I want to recommend shortening – the details may turn out to be important for readers according to their specific interests.The discussion of the conserved and diverged aspects may profit from shortening (something like 20%?).

We have now shortened the discussion about the noduli, which was disproportionately long.

The only typo I found:Line 299 …entirely absent PEN neurons …. did you mean "devoid of"?

Corrected